# A high-throughput platform for single-molecule tracking identifies drug interaction and cellular mechanisms

David Trombley McSwiggen[1], Helen Liu[1], Ruensern Tan[1], Sebastia Agramunt Puig[1], Lakshmi B Akella[1], Russell Berman[1], Mason Bretan[1], Hanzhe Chen[1], Xavier Darzacq[1,2], Kelsey Ford[1], Ruth Godbey[1], Eric Gonzalez[1], Adi Hanuka[1], Alec Heckert[1], Jaclyn J Ho[1], Stephanie L Johnson[1], Reed Kelso[1], Aaron Klammer[1], Ruchira Krishnamurthy[1], Jifu Li[1], Kevin Lin[1], Brian Margolin[1], Patrick McNamara[1], Laurence Meyer[1], Sarah E Pierce[1], Akshay Sule[1], Connor Stashko[1], Yangzhong Tang[1], Daniel J Anderson[1], Hilary P Beck[1]*

[1]Eikon Therapeutics Inc, Hayward, United States; [2]University of California, Berkeley, Berkeley, United States

## eLife assessment

This work presents an **important** technological advance, in the form of a high throughput platform for Single Particle Tracking allowing us to measure millions of cells and thousands of compounds per day. Analysis of the diffusional behaviour of fluorescently-tagged targets permits the identification of, and differentiation between, small molecules that bind directly or affect the target indirectly. The methodology and metrics employed are **compelling**, leading to the identification of multiple compounds that effectively change the diffusive state of the estrogen receptor, the POC target of the study.

*For correspondence:
beckh@eikontx.com

**Abstract** The regulation of cell physiology depends largely upon interactions of functionally distinct proteins and cellular components. These interactions may be transient or long-lived, but often affect protein motion. Measurement of protein dynamics within a cellular environment, particularly while perturbing protein function with small molecules, may enable dissection of key interactions and facilitate drug discovery; however, current approaches are limited by throughput with respect to data acquisition and analysis. As a result, studies using super-resolution imaging are typically drawing conclusions from tens of cells and a few experimental conditions tested. We addressed these limitations by developing a high-throughput single-molecule tracking (htSMT) platform for pharmacologic dissection of protein dynamics in living cells at an unprecedented scale (capable of imaging $>10^6$ cells/day and screening $>10^4$ compounds). We applied htSMT to measure the cellular dynamics of fluorescently tagged estrogen receptor (ER) and screened a diverse library to identify small molecules that perturbed ER function in real time. With this one experimental modality, we determined the potency, pathway selectivity, target engagement, and mechanism of action for identified hits. Kinetic htSMT experiments were capable of distinguishing between on-target and on-pathway modulators of ER signaling. Integrated pathway analysis recapitulated the network of known ER interaction partners and suggested potentially novel, kinase-mediated regulatory mechanisms. The sensitivity of htSMT revealed a new correlation between ER dynamics and the ability of ER antagonists to suppress cancer cell growth. Therefore, measuring protein motion at scale is a powerful method to investigate dynamic interactions among proteins and may facilitate the identification and characterization of novel therapeutics.

## Introduction

Both the activity and mobility of proteins within the crowded cellular environment are profoundly influenced by interactions with their surroundings (*Heo et al., 2022*; *Guin and Gruebele, 2019*; *Shen et al., 2017*). Under these conditions, where diffusion is no longer well described by the Stokes radius of the protein monomer (*Skóra et al., 2020*), changes in protein motion might be expected to correlate closely with changes in the activity of these proteins. The development of increasingly sophisticated live-cell microscopy techniques, including early ensemble methods like fluorescence recovery after photobleaching (FRAP) and fluctuation correlation spectroscopy, have informed our understanding of protein dynamics in cellular biology (*Liu et al., 2015*). A myriad of technical improvements, such as enhanced labeling methods (*Los et al., 2008*; *Mollwitz et al., 2012*), better live-cell compatible fluorophores (*Grimm et al., 2015*), new forms of light microscopy (*Tokunaga et al., 2008*), dramatic increases in computational power (*Chenouard et al., 2014*), and the addition of machine learning approaches to data analysis *Chenouard et al., 2014*; *Speiser et al., 2021* have together enabled a new era of imaging-based studies across biological contexts (*Shen et al., 2017*; *Liu et al., 2015*; *Boka et al., 2021*).

Applying any microscopy technique at scale presents challenges, but recent advances have shown the power of high content imaging techniques to address both mechanistic biological questions as well as to generate leads for new chemical matter in drug discovery (*Chandrasekaran et al., 2021*). Even these conceptually simple experiments involving fixed cells stained with well-characterized commercial reagents take careful experiment design and sophisticated computational approaches to execute (*Caicedo et al., 2017*). It is no wonder, then, that attempts to combine high content imaging workflows with more advanced super-resolution microscopy methods have thus far been limited. Such advances have enabled the development of systems for fixed-cell STORM imaging at an impressive 10,000 cells/day (*Barentine et al., 2023*), though the appropriate application of this increase in scale remains an open question.

In single-molecule tracking (SMT), a fluorescently labeled protein of interest is imaged at high spatiotemporal resolution to track its motion in a live cell (*Chenouard et al., 2014*). The information embedded in these trajectories has been used to investigate diverse cellular phenomena including protein oligomerization state and function (*Barentine et al., 2023*; *Needham et al., 2016*; *Yasui et al., 2018*), inter-organelle communication (*Nixon-Abell et al., 2016*), nuclear organization (*Hansen et al., 2017*), and transcription regulation (*Paakinaho et al., 2017*; *Mazza et al., 2012*; *Swinstead et al., 2016*; *Presman et al., 2017*). Of particular utility are 'fast-SMT' approaches which use high frame rates and stroboscopic illumination to minimize motion-induced blurring, and hence can measure diffusive states over a large dynamic range (*Izeddin et al., 2014*; *Hansen et al., 2018*). Specifically, proteins that diffuse rapidly throughout the cell are often missed in alternative tracking approaches, biasing the resulting data. In spite of the potential biological discoveries that depend on the application of SMT on a large scale, SMT in general (and fast-SMT in particular) has not been adapted to a high-throughput setting that would enable the analysis of complex, multi-component systems, or the identification of compounds that affect protein motion.

Steroid hormone receptors (SHRs) are a class of transcription factors that play crucial roles in normal human development and in disease pathogenesis. SHRs like the estrogen receptor (genes ESR1 and ESR2), androgen receptor (AR), and progesterone receptor (PR), as examples, contribute decisively to the acquisition of secondary sex characteristics, while the glucocorticoid receptor (GR) helps to orchestrate both metabolism and inflammation (*Ahmad and Kumar, 2011*). In their ligand-free state, SHRs are kept sequestered in multiprotein complexes by the chaperone HSP90 (*Saha et al., 2021*). Canonically, in the presence of hormone they dimerize and bind their cognate genomic response elements, recruiting epigenetic modifiers and transcription machinery (*Ahmad and Kumar, 2011*; *Saha et al., 2021*; *Papageorgiou et al., 2021*). At the same time, SHR-derived signals impose a large disease burden by promoting the growth of breast cancers (ER) (*Ahmad and Kumar, 2011*; *Lu and Liu, 2020*) or prostate cancers (AR) *Ahmad and Kumar, 2011*; or by imposing immune and metabolic dysfunction (GR) *Ahmad and Kumar, 2011*; *Quatrini and Ugolini, 2021*. SHRs therefore provide an excellent proof-of-concept system to study the relationship between protein dynamics and protein function due to the wealth of information and reagents (*Lu and Liu, 2020*) already available for these systems as well as previous reports characterizing some aspects of their cellular dynamics (*Paakinaho et al., 2017*; *Swinstead et al., 2016*; *Guan et al., 2019*; *Van Royen et al., 2014*; *Van Royen et al., 2012*).

Here, we present the first industrial-scale, high-throughput, fast-SMT (htSMT) platform capable of measuring protein motion from more than 13,000 individual assay wells (>1,000,000 individual cells) per day. Using ER as a test system, we demonstrate that chemical screening using htSMT is specific, robust, and reproducible. The increase in throughput enables classical drug discovery activities, including compound library screening and the elucidation of structure–activity relationships (SARs), yielding accurate and reproducible results that are inaccessible or unmeasurable with other techniques or using SMT on a smaller scale. Importantly, we demonstrate that htSMT can be used to characterize both known and novel pathway contributions to the ER protein interaction network. More than a proof-of-concept for the htSMT platform, these data confirm that analysis of protein motion itself on a large scale reveals detailed information about pathway interactions and signaling.

## Results
### Creation and validation of a high-throughput SMT platform

We developed a robotic system capable of handling reagents, collecting high-quality fast-SMT image series, processing time-ordered raw images to yield molecular trajectories, and extracting features of biological interest within defined cellular compartments (*Figure 1—figure supplement 1A, B*). Samples start as cells seeded into 384-well plates in a hotel incubator. A central robotic arm retrieves the plates, and delivers them to an Echo 650 acoustic dispenser to add dye. After incubating, excess dye is washed away and Echo 650 is again used to administer compound treatment. Stained and compound-treated plates are then delivered to any of up to four identical SMT microscopes for imaging. Both SMT and accompanying Hoechst images are collected and automatically processed to identify individual molecule positions, reconnect the spot coordinates into trajectories, and then associate each trajectory with a nuclear mask. Finally, the processed SMT data are subjected to quality control to omit aberrant fields of view using a convolutional neural network trained to identify technical errors in the images (*Figure 1—figure supplement 1C*), and finally stored for downstream analysis.

To examine htSMT system performance across a broad spectrum of diffusion coefficients, we generated three U2OS cell lines ectopically expressing HaloTag (*Los et al., 2008*) fused proteins with well-established behaviors in the cell. These HaloTag fusions allow the subsequent addition of bright and photostable organic fluorophores like JF$_{549}$ (*Grimm et al., 2015*) which produce high signal spots to detect and track. Histone H2B-Halo, which is predominantly incorporated into chromatin and therefore effectively immobile over short timescales (*Hansen et al., 2018*), was employed to estimate localization error. A prenylation motif (Halo-CaaX) embedded in the plasma membrane exhibits moderate diffusion (*Natwick and Collins, 2021*). Unfused HaloTag was chosen to represent the upper limit of cellular 'free' diffusion. Single-molecule trajectories measured in these cell lines yielded diffusion coefficients for CaaX similar to published results (*Natwick and Collins, 2021*), and the diffusion for H2B-HaloTag was consistent with the theoretical lower bounds that can be approximated from the localization error and 10-ms frame interval ($D_{\text{apparent}} = D_{\text{true}} + \frac{\text{localization error}^2}{\Delta\tau}$) (*Figure 1A*). Localization error can be measured directly from the single-molecule trajectories using the jump covariance of slow or immobile particles (*Heckert et al., 2022*). Using the immobile H2B-Halo trajectories, we found the localization error of the htSMT system to be 39 nm (*Figure 1—figure supplement 2A*), comparable to other benchmark stroboscopic illumination datasets (*Hansen et al., 2018*; *Heckert et al., 2022*). The diffusion coefficient for free HaloTag is consistent with previous SMT reports (*Hansen et al., 2018*), but is also within the theoretical upper bounds of a Mean Squared Displacement (MSD) estimator of diffusion coefficient for a 10-ms frame interval and 1.25 µm search radius ($D_{\text{apparent}} \leq \frac{R_{\text{search}}^2}{8\Delta\tau}$), thus we consider the distribution recovered from the free HaloTag to represent the upper limit of trackable particles with this assay configuration.

We then tested whether our htSMT platform can extract accurate molecular trajectories at scale. We employed 384-well plates where free Halo, Halo-CaaX, and H2B-Halo cell lines were mixed in equal proportions in each well. Imaging with a 94 µm by 94 µm field of view (FOV), we achieved an average of 10 nuclei simultaneously (*Figure 1B*, *Figure 1—figure supplement 2B*, *Figure 1—video 1*), enough that most FOVs contained cells from each cell line. To limit ambiguity in cell assignment, we considered only the trajectories that fell within nuclear segmented regions. The probability distribution of diffusion states cleanly distinguishes between the three cell types (*Figure 1C*). More importantly, by looking at the single-cell state distribution profiles of 103,757 cells from five separate

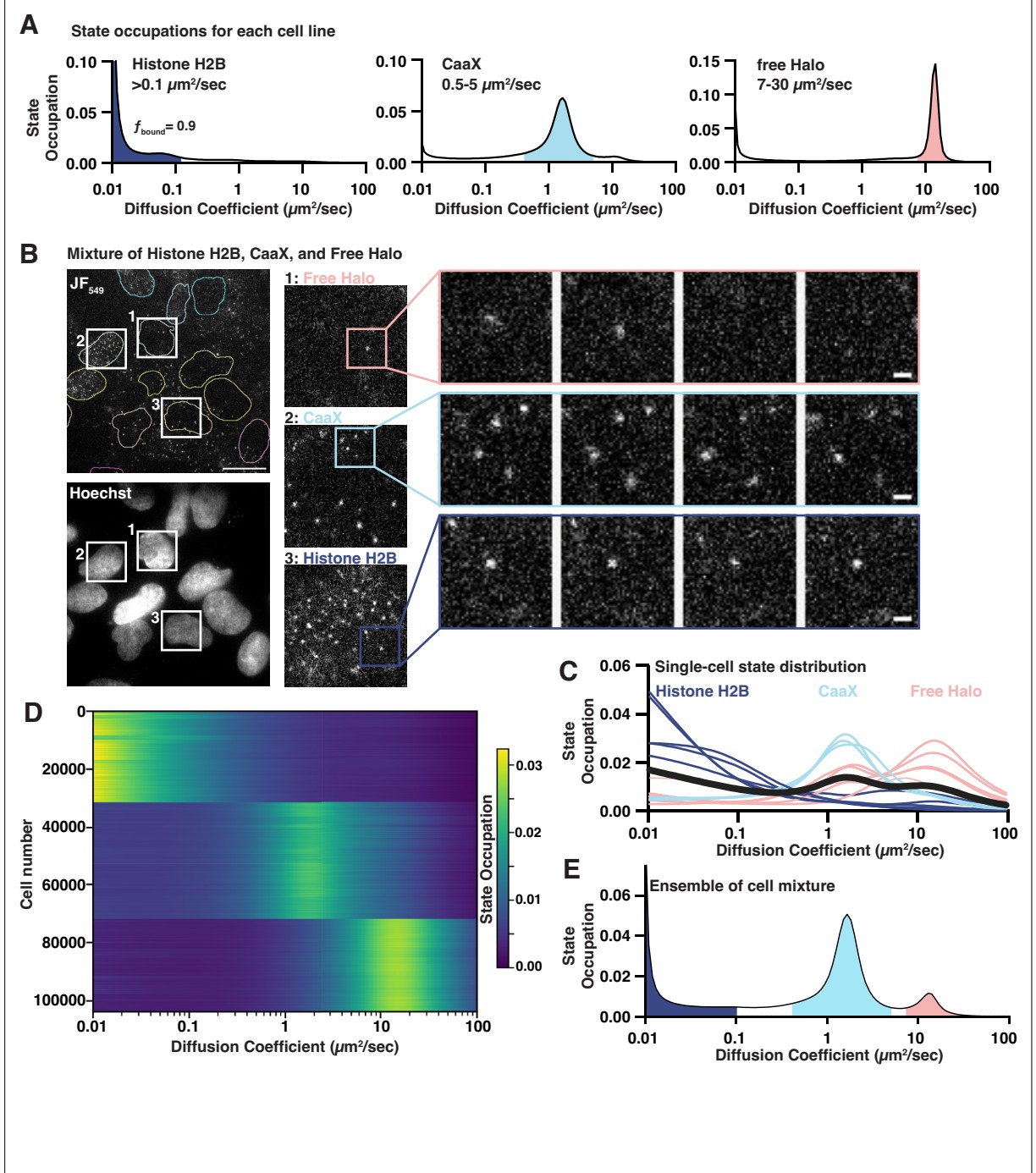

**Figure 1.** Benchmarking a high-throughput single-molecule tracking platform. (**A**) Diffusion state probability distributions from three cell lines expressing Histone H2B-Halo, Halo-CaaX, or Halo alone. Shaded bins represent the diffusive states characteristic of each cell line. (**B**) Example field of view. Equal mixture of H2B-Halo, Halo-CaaX, and free Halo cell line single-molecule images (top) and reference Hoechst image (bottom). Insets show zoom-ins to individual cells and to sequential frames of individual molecules. Image intensities are equivalently scaled across panels. (**C**) Single-cell diffusive states extracted from, colored based on similarity to the H2B-, CaaX-, or free-Halo dynamics reported in (**B**). (**D**) Heatmap representation of 103757 cell nuclei measured from a mixture of Halo-H2B, Halo-CaaX, or free Halo mixed within each well over 1540 unique wells in five 384-well plates. Each horizontal line represents a nucleus. Cells were clustered using *k*-means clustering and labels assigned based on the diffusive profiles determined in (**B**). (**E**) Ensemble state occupation of all trajectories recovered from a mixture of Halo-H2B, Halo-CaaX, and free Halo cells. Mean state occupation from 308 assay wells.

The online version of this article includes the following video, source data, and figure supplement(s) for figure 1:

**Source data 1.** TIFF versions of the images in *Figure 1*.

*Figure 1 continued on next page*

*Figure 1 continued*

**Figure supplement 1.** Overview of the high-throughput single-molecule tracking (htSMT) workflow.

**Figure supplement 2.** Characterization of high-throughput single-molecule tracking (htSMT) system performance.

**Figure 1—video 1.** Example SMT field of view as shown in *Figure 1*.

https://elifesciences.org/articles/93183/figures#fig1video1

384-well plates, grouped by their distribution profile, we recovered highly consistent estimates of protein dynamics at the single cell level, comparable to the pure populations (*Figure 1D*, *Figure 1— figure supplement 2C*).

While single-cell measurements are powerful, the number of trajectories in one cell is limited, and so estimates of diffusive states can be broad. Combining trajectories from multiple cells, however, provides the expected distribution of diffusive states (*Figure 1E*). Moreover, combining trajectories derived from many cells makes model-based (*Hansen et al., 2018*) or model-agnostic (*Heckert et al., 2022*) state analysis possible, where as few as $10^3$ trajectories permit satisfactory inference of the underlying diffusion states. We determined that imaging six fields of view (FOV), for 1.5 s each, yielded enough trajectories (>10,000) to accurately estimate protein dynamics, bringing the overall throughput of the platform to more than 13,000 individual wells (>90,000 FOVs; >1,000,000 cells/ day), a rate of data acquisition that enables compound screening on a feasible timescale (*Figure 1— figure supplement 2D*).

## Using htSMT to measure protein dynamics of SHRs

Equipped with an htSMT system capable of measuring protein dynamics broadly, we sought to understand how measuring protein motion can be used to characterize protein activity. SHRs transition between inactive and active states via ligand binding (*Figure 2—figure supplement 1A*), a phenomenon that has been previously observed at the single-molecule level (*Paakinaho et al., 2017*; *Swinstead et al., 2016*), and we hypothesized that the large dynamic range and orders-of-magnitude increase in throughput of our platform could capture these differences in the context of compound screening. We generated HaloTag fusions to ER, AR, and PR through ectopic expression, and GR through endogenous knock-in. Similar to previous approaches (*Paakinaho et al., 2017*; *Wagh et al., 2023*), we used clonal cell lines in a U2OS cell background to minimize effects of comparing dynamics in different cell types. Clones were carefully selected such that the HaloTag fusion SHRs were comparable to each other in transcript abundance, and not higher than transcript levels in tissue-specific cell lines like MCF7 and T47d, which are both ER and PR positive (*Figure 2—figure supplement 1B*).

In the absence of hormone, all four SHR proteins exhibit similar dynamic profiles: a small immobile fraction and a large freely diffusing fraction with a 3.4–4.3 $\mu m^2$/s average diffusion coefficient (*Figure 2A*, *Figure 2—figure supplement 1C*). No correlation between diffusion and protein molecular weight (138 kDa for Halo-AR, 102 kDa for ER-Halo, 122 kDa for Halo-GR, and 135 kDa for Halo-PR) was observed, highlighting the differences between cellular protein dynamics versus purified systems. Upon addition of agonist, a dramatic increase in immobile trajectories is observed, which we attribute to chromatin binding. Using a conservative upper-bound of chromatin mobility in the nucleus and chromatin-associated transcription factors (*Heckert et al., 2022*), we define the bound fraction ($f_{bound}$) for each SHR as the fraction of trajectories diffusing less than 0.1 $\mu m^2$/s (*Figure 2A*). Using this threshold, $f_{bound}$ of histone H2B is 0.92 on average, consistent with previous reports (*Hansen et al., 2018*). $D_{free}$ we defined as the occupation-weighted average diffusion coefficient of the non-bound states (*Figure 2—figure supplement 1C*). Some SHRs had a higher proportion of bound molecules than others. The ligand-induced effect is most pronounced for ER, with 34% bound in basal conditions and 87% bound after estradiol treatment (*Figure 2A*, *Videos 1 and 2*).

SHRs are highly selective for their cognate agonists in biochemical binding assays, which we confirmed by measuring the dose-dependent change in dynamics as a function of agonist concentration. The maximal change in $f_{bound}$ (*Figure 2B*) and decrease in $D_{free}$ (*Figure 2—figure supplement 1D*) differed between SHRs. The dose titration curves also showed variable potencies ($EC_{50}$) for each SHR/hormone pair, with ER-estradiol being both the most potent and most selective pair. RNA-seq after estradiol stimulation showed a marked induction of hallmark ER-dependent gene sets (*Liberzon et al., 2015*), confirming that the increase in chromatin binding we observed by SMT has a

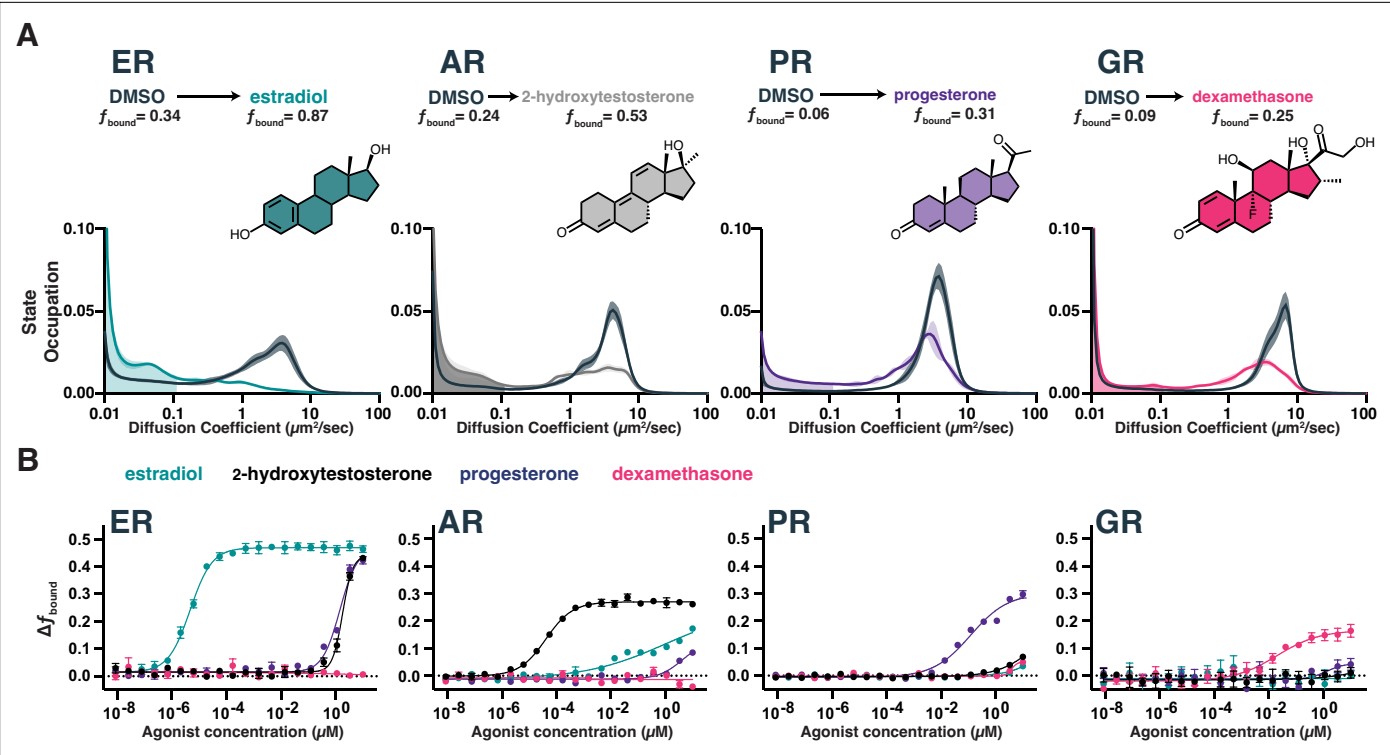

**Figure 2.** Chromatin binding by steroid hormone receptors (SHRs) is affected by compound treatment. (**A**) Distribution of diffusive states for Halo-AR, Halo-ER, Halo-GR, and Halo-PR in U2OS cells before and after stimulation with an activating ligand. The area in the shaded region is $f_{bound}$. Shaded error bands represent SD. (**B**) Selectivity of individual SHRs to their cognate ligand compared with other steroids, as determined by $f_{bound}$. Error bars represent SEM across three biological replicates.

The online version of this article includes the following source data and figure supplement(s) for figure 2:

**Source data 1.** Tabular data to generate plots from *Figure 2* and related figure supplements.

**Figure supplement 1.** Activation of steroid hormone receptors (SHRs) changes free diffusion and impacts downstream gene expression.

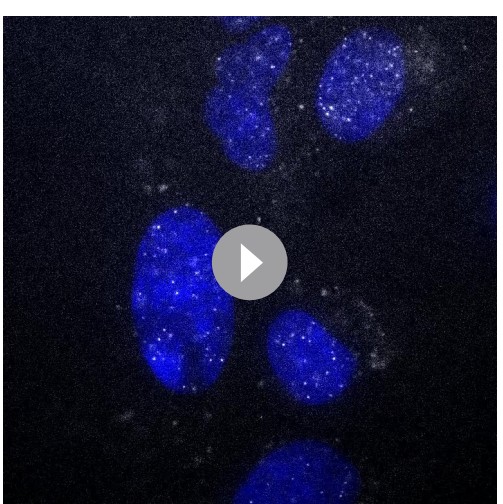

**Video 1.** Example field of view of single-molecule tracking (SMT) from Halo-ER cells. Video is played back at 10 frames per second, 10× slower than real time.
https://elifesciences.org/articles/93183/figures#video1

functional effect in promoting ER-responsive gene programs, even in the ectopic expression setting (*Figure 2—figure supplement 1E, F*). Thus, SMT can detect functionally relevant changes in transcription factor dynamics and accurately differentiate the ligand/target specificity directly within the cellular environment.

## Screening a diverse bioactive chemical set identifies known and novel modulators of ER dynamics

Our characterization efforts of ligand selectivity for AR, ER, GR, and PR collectively suggested that we could use SMT to interrogate the effects of compounds on protein dynamics at a throughput conducive to high-throughput screening. We first identified a structurally diverse set of 5067 molecules with heterogeneous biological activities as a useful screening set. Having determined that the same data acquisition parameters (6 FOVs imaged for 1.5 s each) were sufficient to recover

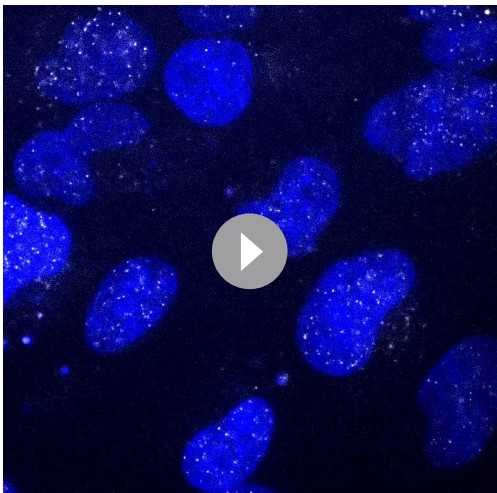

**Video 2.** Example field of view of single-molecule tracking (SMT) from Halo-ER cells treated with 25 nM estradiol for 1 hr prior to imaging. Video is played back at 10 frames per second, 10× slower than real time.
https://elifesciences.org/articles/93183/figures#video2

more than 10,000 trajectories per well, we could achieve a throughput of 15,000 wells per day which would permit us to interrogate the whole library in a single day (*Figure 3—figure supplement 1A*). For cells at steady state, such a brief acquisition time also means that the dynamical state of the cell remains largely constant within an FOV (*Figure 3—figure supplement 1B*), allowing all of the trajectories from an FOV to be considered largely representative of the same cellular state. We chose to screen this bioactive compound set against ER, assessing change in $f_{bound}$ at 1 µM compound versus the vehicle dimethyl sulfoxide (DMSO) (*Figure 3*, *Figure 3—figure supplement 1C, D*, *Figure 3—source data 1*). The screen was run twice to assess reproducibility, showing a high degree of agreement between replicates for ER-active molecules (*Figure 3—figure supplement 2A*). This screen illustrates some important advantages of our htSMT platform over more manual lower-throughput approaches.

From plate to plate, the assay window for the screen was robust (*Zhang et al., 1999*) (average $Z'$-factor = 0.79 over 72 plates), the measured potency of the control estradiol in each instance remained within threefold of the mean (*Figure 3—figure supplement 2B–D*), and the distribution of negative control wells centered tightly on zero (*Figure 3—figure supplement 2E, F*). Each compound measurement was averaged from SMT trajectories of between 94 and 161 cells (25th to 75th percentiles; *Figure 3—figure supplement 2G*). Of the 30 compounds, we identified from the bioactive set

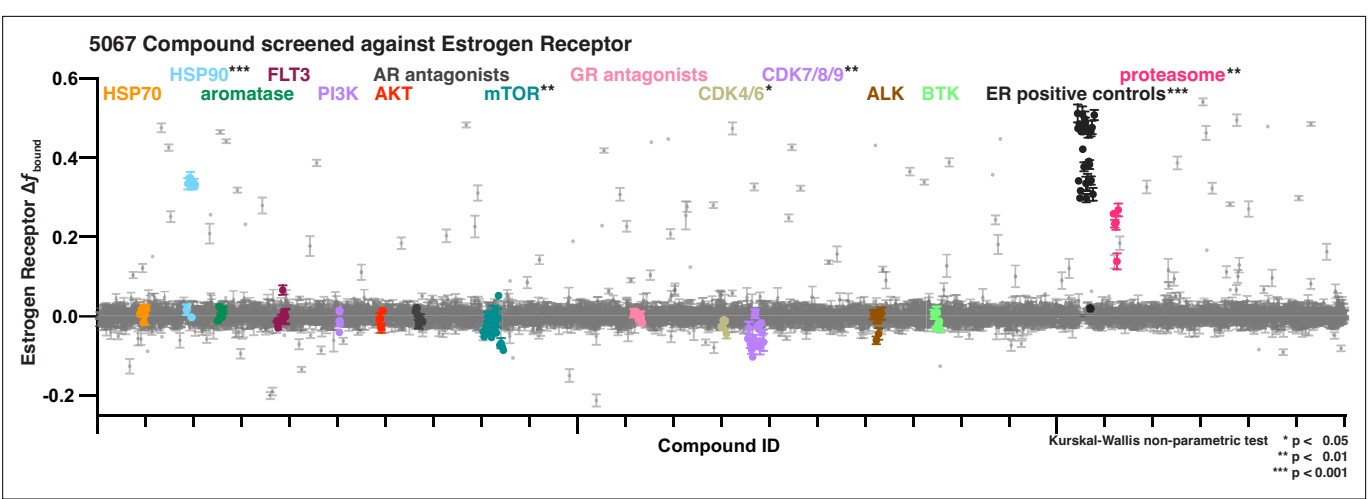

**Figure 3.** Bioactive screen results of change in estrogen receptor $f_{bound}$ for two biological replicates, each with 2 well replicates per compound of 5067 compounds. Select inhibitors are grouped and uniquely colored by pathway or target. Error bars are SEM of all replicates.

The online version of this article includes the following source data and figure supplement(s) for figure 3:

**Source data 1.** List of compounds as part of the Sellek L-1700 bioactive set along with the 'Target' and 'Pathway' annotations as identified by Selleck.

**Source data 2.** Tabular data to generate plots from *Figure 3* and related figure supplements.

**Figure supplement 1.** Experimental design for bioactive molecule screen.

**Figure supplement 2.** Screen of bioactive molecules produces robust data with good assay performance.

**Figure supplement 3.** Contributions of different sources of variability of jump length in the high-throughput single-molecule tracking (htSMT) bioactive compound screen against ER.

**Figure supplement 4.** Antagonists of estrogen receptor (ER).

that we expected to modulate ER, either as agonists or antagonists (*Lu and Liu, 2020*; *Kuiper et al., 1997*), all significantly increased $f_{bound}$ measured by SMT. This includes agonistic molecules like estradiol, but also notable examples such as 4-hydroxytamoxifen (4-OHT), fulvestrant, and bazedoxifene (*Figure 3*, Extended Data 3B, H).

With a large dataset to work from, collected across multiple assay plates on multiple independent microscopes, we examined the sources of variability within the screening assay. Using random sampling of individual jumps within the screening dataset while holding constant the source of the jumps (e.g. sampling jumps within a specific assay well), we could estimate the relative contributions of microscope-to-microscope, plate-to-plate, well-to-well, FOV-to-FOV, and cell-to-cell variability. Cell-to-cell variability was the single largest contributor to overall assay variability, especially when considering only vehicle treated controls (*Figure 3—figure supplement 3B*), and stabilized after ~1000 jumps (*Figure 3—figure supplement 3C*). These results support the use of a short acquisition time per FOV and multiple FOVs to stabilize the dynamical state estimate.

The somewhat counter-intuitive finding that either strong agonism or antagonism can lead to an increase in chromatin binding has been reported for ER (*Guan et al., 2019*), but this appears not to a general feature of SHRs. While the PR antagonist mifepristone (*Goyeneche et al., 2007*) behaves similarly to ER (*Figure 3—figure supplement 4A*), antagonists of AR like Enzalutamide and Darolutamide (*Rajaram et al., 2020*), and antagonists of GR like AL082D06 (*Miner and Tyree, 2003*) cause a decrease in chromatin binding. This decrease occurs when administered singly or when co-administered in competition with the cognate agonist (*Figure 3—figure supplement 4B, C*). These results show how the cellular context and interaction partners are critical to understand the effect of a compound on its intended target. To underscore this point, in addition to binders of the ER ligand-binding domain, we also identified a number of active compounds targeting diverse nodes in the ER interaction network, including modulators of the proteasome, chaperones, kinases, and others (*Figure 3*).

## Cellular ER dynamics elucidate SARs of ER modulators

Our screen revealed that, surprisingly, all the known ER modulators—both agonists like estradiol and potent antagonists like fulvestrant—caused an increase in $f_{bound}$ (*Figure 4A*). We therefore characterized a subset of selective ER modulators (SERMs) and selective ER degraders (SERDs) in more detail. These molecules all bind competitively to the ER ligand binding domain (*Lu and Liu, 2020*). As in the bioactive screen, both SERDs and SERMS increased $f_{bound}$ (*Figure 4A*) and slightly decreased measured $D_{free}$ (*Figure 4—figure supplement 1A*), with potencies ranging from 9 pM for GDC-0927 to 4.8 nM for GDC-0810 (*Figure 4B*). To understand how quickly these compound effects take place, we measured the change in $f_{bound}$ as a function of time, collecting timepoints roughly every 2 min beginning immediately after compound addition. Despite different physical–chemical properties, all five increased $f_{bound}$ within minutes of compound addition (*Figure 4C*, *Figure 4—figure supplement 1B*), with no evidence of transient states distinct from the free diffusion and chromatin bound peaks (*Figure 4—figure supplement 1C*). Presumably, ER dissociation from the chaperone complex, dimerization, and chromatin binding occur on rapid and seemingly comparable timescales. Since we cannot distinguish individual steps in these transitions, we consider the on rate of the entire process to have the effective rate constant $k^*_{on}$. Importantly, selective antagonists of AR and GR did not induce significant modulation of ER dynamics, further highlighting the utility of htSMT in characterizing the specificity of interactions between small molecule modulators of protein function and their cognate targets (*Figure 4—figure supplement 1D*).

Interestingly, SERMs 4-OHT and GDC-0810 show lower maximal increases in $f_{bound}$ compared with the SERDs fulvestrant and GDC-0927 (*Figure 4D*). Similar effects have been described previously using FRAP (*Guan et al., 2019*), which we confirmed using our Halo-ER cell line (*Figure 4E*). The delay in ER signal recovery after 2 min in FRAP was consistent with the changes in $f_{bound}$ measured by SMT (*Figure 4F*). Although FRAP was used to measure these $f_{bound}$ differences, the technique suffers from challenges in scalability and depends heavily on prior assumption of the underlying dynamical model in the sample. SMT, alternatively, enables detailed characterization of the potency of 4-OHT and GDC-0810 relative to other ER ligands in their ability to increase ER chromatin binding on a tractable timescale.

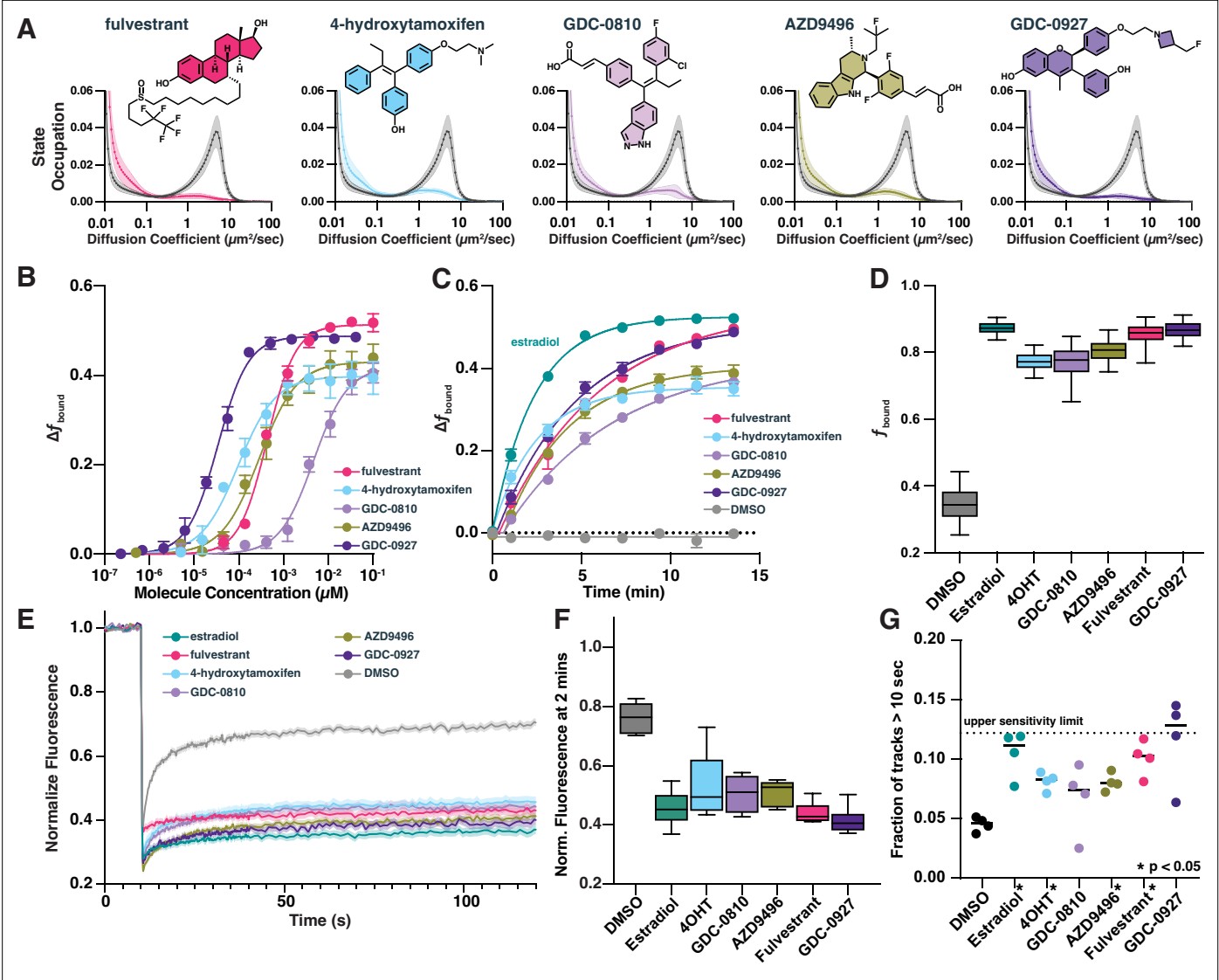

**Figure 4.** Selective estrogen receptor (ER) modulators and degraders induce DNA-binding measurable through high-throughput single-molecule tracking (htSMT). (**A**) Diffusion state probability distribution for ER treated with 100 nM of exemplified selective ER modulators (SERMs) and selective ER degraders (SERDs). Each state distribution is generated from 10,000 randomly sampled nuclear trajectories per assay well. Shaded regions represent the SD. (**B**) Change in $f_{bound}$ as a function of a 12-point dose titration of fulvestrant (5), 4-hydroxytamoxifen (4-OHT) (6), GDC-0810 (8), AZD9496 (9), or GDC-0927 (10) with fitted curve. Compounds colored as in (A). Error bars represent the SEM across three biological replicates. (**C**) Change in $f_{bound}$ as a function of time after agonist or antagonist addition, fitted with a single exponential. Compounds colored as in (A). Estradiol (1, green) and DMSO added for comparison. Error bars represent SEM across three biological replicates. (**D**) Maximum effect of SERMs and SERDs on $f_{bound}$. Each box represents quartiles while whiskers denote the 5–95th percentiles of single well measurements, measured over a minimum of 4 days with at least 8 wells per compound per day. (**E**) Fluorescence recovery after photobleaching (FRAP) of ER-Halo cells, treated either with DMSO alone or with 100 nM SERM/D. Curves are the mean ± SEM for 18–24 cells, colored as in (A). (**F**) Quantification of FRAP recovery curves to measure recovery 2 min after photobleaching. Whiskers denote the 5–95th percentiles of single cell measurements. (**G**) Quantification long-lived tracks, each point representing the fraction of trajectories greater than 10 s for a single biological replicate consisting of 3–10 wells per condition. Dashed line represents the median fraction of trajectories lasting longer than 10 s for Histone H2B-Halo, which is the upper limit of measurement sensitivity. * indicates sample with $p < 0.05$ as measured by $t$-test.

The online version of this article includes the following source data and figure supplement(s) for figure 4:

**Source data 1.** Tabular data to generate plots from *Figure 4* and related figure supplements.

**Figure supplement 1.** Selective estrogen receptor modulators (SERMs) and selective estrogen receptor degraders (SERDs) decrease free diffusion and increase $f_{bound}$ rapidly after addition.

Neither FRAP nor htSMT can discriminate between recovery driven by an increase in residence time (decreasing $k^*_{off}$) or increasing the rate of chromatin binding (increasing $k^*_{on}$), either of which would result in increasing $f_{bound}$. By changing SMT acquisition conditions to reduce the illumination intensity and collect long, 250 ms continuous frame exposures, only immobile proteins form spots. Under these imaging conditions, the distribution of track lengths provides a measure of relative residence times (*Paakinaho et al., 2017*; *Mazza et al., 2012*; *Liu et al., 2014*; *Elf et al., 2007*). Both agonist and antagonist treatment led to longer binding times compared to DMSO, suggesting that ligand binding decreases $k^*_{off}$ (*Figure 4G*, *Figure 4—figure supplement 1E*). Consistent with FRAP, estradiol, GDC-0927, and fulvestrant show longer binding times compared with other ER modulators. Using $f_{bound}$ and $k^*_{off}$ measurements, one can infer the $k^*_{on}$. In all cases, the changes in dissociation rate are not proportional to the increase in $f_{bound}$, and so ligand-imposed increases in $k^*_{on}$ likely contribute to the observed change in the chromatin-associated ER fraction (*Figure 4—figure supplement 1F*). These data are consistent with a model wherein ER rapidly binds to chromatin irrespective of which molecule occupies the ligand-binding domain, but some ligands induce a conformation that can be further stabilized on chromatin by co-factors. Consequently, these data support the hypothesis that ER may engage chromatin in mechanistically different ways (*Guan et al., 2019*). An efficacious ER inhibitor may promote rapid and transient chromatin binding that fails to effectively recruit necessary co-factors to drive transcription (*Van Royen et al., 2012*).

## htSMT reveals a relationship between ER dynamics and efficacy in ER-dependent cell toxicity

As the name implies, next-generation ER degraders like GDC-0927, AZD9833, and GDC9545 were optimized to enhance degradation of ER (*Guan et al., 2019*; *Chen et al., 2022*). We indeed observed compound-induced ER degradation via immunofluorescence, both in established breast cancer model lines and our U2OS ectopic expression system (*Figure 5—figure supplement 1A, B*). Structural analogs of GDC0927 have been reported and optimized for ER degradation, however the

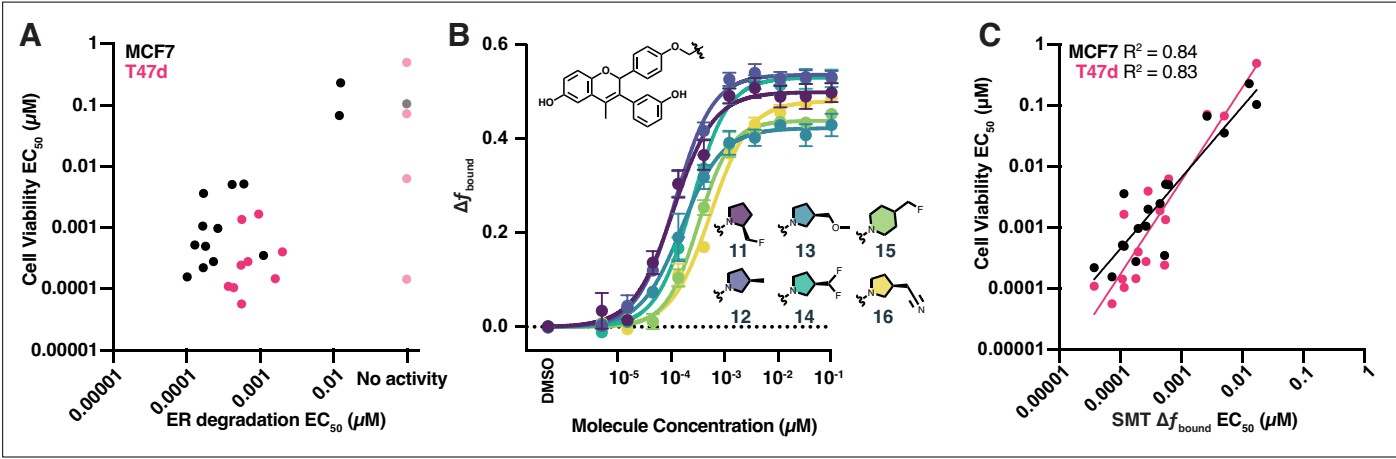

**Figure 5.** High-throughput single-molecule tracking (htSMT) can be used to determine chemical structure–activity relationships. (**A**) Correlation of potency measured by estrogen receptor (ER) degradation and cell proliferation in MCF7 cells (black) and T47d cells (magenta) for compounds in the GDC-0927 structural series. Cell proliferation and ER degradation-derived potencies are the mean of four and three biological replicates, respectively. (**B**) Change in $f_{bound}$ across a 12-point dose titration of compounds 11 through 16, colored by structure. Points are the mean ± SEM across three biological replicates. (**C**) Correlation of potency measured by change in $f_{bound}$ and cell proliferation in MCF7 cells (black) and T47d cells (magenta) for compounds in the GDC-0927 structural series. Cell proliferation and SMT-derived potencies are the mean of four and three biological replicates, respectively.

The online version of this article includes the following source data and figure supplement(s) for figure 5:

**Source data 1.** PDF report of western blot produced by Jess system (Protein Simple).

**Source data 2.** TIFF versions of the images in *Figure 5*.

**Source data 3.** Images to generate plots from *Figure 5* and related figure supplements.

**Source data 4.** Tabular data to generate plots from *Figure 5* and related figure supplements.

**Figure supplement 1.** GDC-0927 structural variants characterized by estrogen receptorER degradation or cell proliferation assays.

correlation between ER degradation and cell proliferation is poor (*Lu and Liu, 2020*; *Kahraman et al., 2019*; *Figure 5A*, *Figure 5—figure supplement 1C–E*). We hypothesized that by measuring protein dynamics we might obtain more precise measurements of inhibitory activity than can be achieved by assessing protein degradation. We therefore determined the potency and maximal effect of structural analogues of GDC-0927 using htSMT.

Overall, these analogues exhibited a potency range of 15 pM to 12 nM and increased ER $f_{bound}$ by 0.4–0.56 (*Figure 5B*). Small changes in the chemical structure produced measurable changes in both compound potency and maximal efficacy as determined using SMT, a critical feature if an assay is to be used to iteratively optimize a compound for potency or efficacy.

We compared the potencies of GDC-0927 and analogues determined either via ER degradation or SMT, with the ability of each of these compounds to block estrogen-induced breast cancer cell proliferation. Potency assessed by ER degradation was not a good predictor of potency in the cell proliferation assay (*Figure 5A*). By contrast, SMT measurements of $f_{bound}$ strongly correlate with cell viability (*Figure 5C*; $R^2$ of 0.83 for T47d and 0.84 for MCF7). Intriguingly, SMT $EC_{50}$ values were on average tenfold lower than those observed in the cell growth assay, suggesting that SMT may be sensitive enough to identify chemical series that would not show effects in other cellular assays, enabling the identification of starting points for medicinal chemistry that could not be obtained by other methods. This correlation between effects on protein dynamics ($f_{bound}$) and protein function (suppression of cell proliferation) coupled with the throughput of the SMT system make this an attractive approach for the identification of protein modulators with novel properties.

## Screening of a diverse chemical library using htSMT enables unbiased pathway interaction analysis by monitoring protein dynamics

In addition to known ER-active modulators, many other compounds in our bioactive library provoked easily measurable changes in $f_{bound}$. To define a threshold for calling a molecule from the screen 'active', we selected 92 compounds with different magnitudes of change in $f_{bound}$ to retest in a dose titration (*Figure 6—figure supplement 1A, B*). We found that a 5% change in $f_{bound}$ was sufficient to reproducibly distinguish active compounds. Using this approach, we identified 239 compounds in the bioactive library that affected the ER mobility (*Figure 6—figure supplement 1B*). Among these compounds, the correlation between the two screen replicates was high ($R^2$ = 0.92) and the level of activity was reproducible (the slope for active molecules was 0.94). Some active compounds could be clustered based on scaffold homology, but most clusters consisted of one or only a few members (*Figure 6—figure supplement 1C, D*). Structural clustering was employed to identify known ER modulators where the vendor-provided annotation was poorly defined (*Figure 6—figure supplement 1C*). Our results demonstrate that htSMT is reproducible and robust when screening large collections of molecules.

Most active molecules from the screen were not structurally related to steroids (*Figure 6—figure supplement 1C, D*). On the other hand, many compounds could be grouped based on their reported biological targets or pathways (*Figure 3*, *Figure 6—figure supplement 3A, B*). For example, heat shock protein (HSP) and proteasome inhibitors consistently increased $f_{bound}$, whereas cyclin-dependent kinase (CDK) and mTOR (mammalian target of rapamycin) inhibitors decreased $f_{bound}$. Though many CDK inhibitors lack within-family specificity (*Wells et al., 2020*; *Figure 6—figure supplement 2A*, pan-CDK), we found that CDK9-specific inhibitors more strongly affected ER dynamics than did CDK4/6-specific inhibitors. Furthermore, as with selective AR and GR antagonists, inhibitors targeting ALK, BTK, and FLT3 kinases that have not been shown to interact with ER have no impact on ER dynamics when assessed using SMT (*Figure 3*).

For the inhibitors of cellular pathways we identified, we used a dose titration to better characterize the effect of each on ER dynamics. Potencies ranged from the sub-nanomolar to low micromolar (*Figure 6B*), similar to the reported potencies of these compounds against their cellular targets (*Wang et al., 2018*; *Olson et al., 2018*; *Parry et al., 2019*; *Menezes et al., 2012*; *McCleese et al., 2009*; *Bussenius et al., 2012*; *Apsel et al., 2008*; *Thoreen et al., 2009*; *Yu et al., 2009*; *Adams et al., 1999*; *Kuhn et al., 2007*; *Mroczkiewicz et al., 2010*). Additionally, we tested these molecules against AR (*Figure 6—figure supplement 2C*) and PR (*Figure 6—figure supplement 2D*). Each SHR differed meaningfully from the others in terms of the response to compounds identified through an ER-focused screening effort. Again, the magnitude of ER SMT effect was largely consistent within a target class (*Figure 6A*, *Figure 6—figure supplement 2A–D*). The finding that structurally distinct

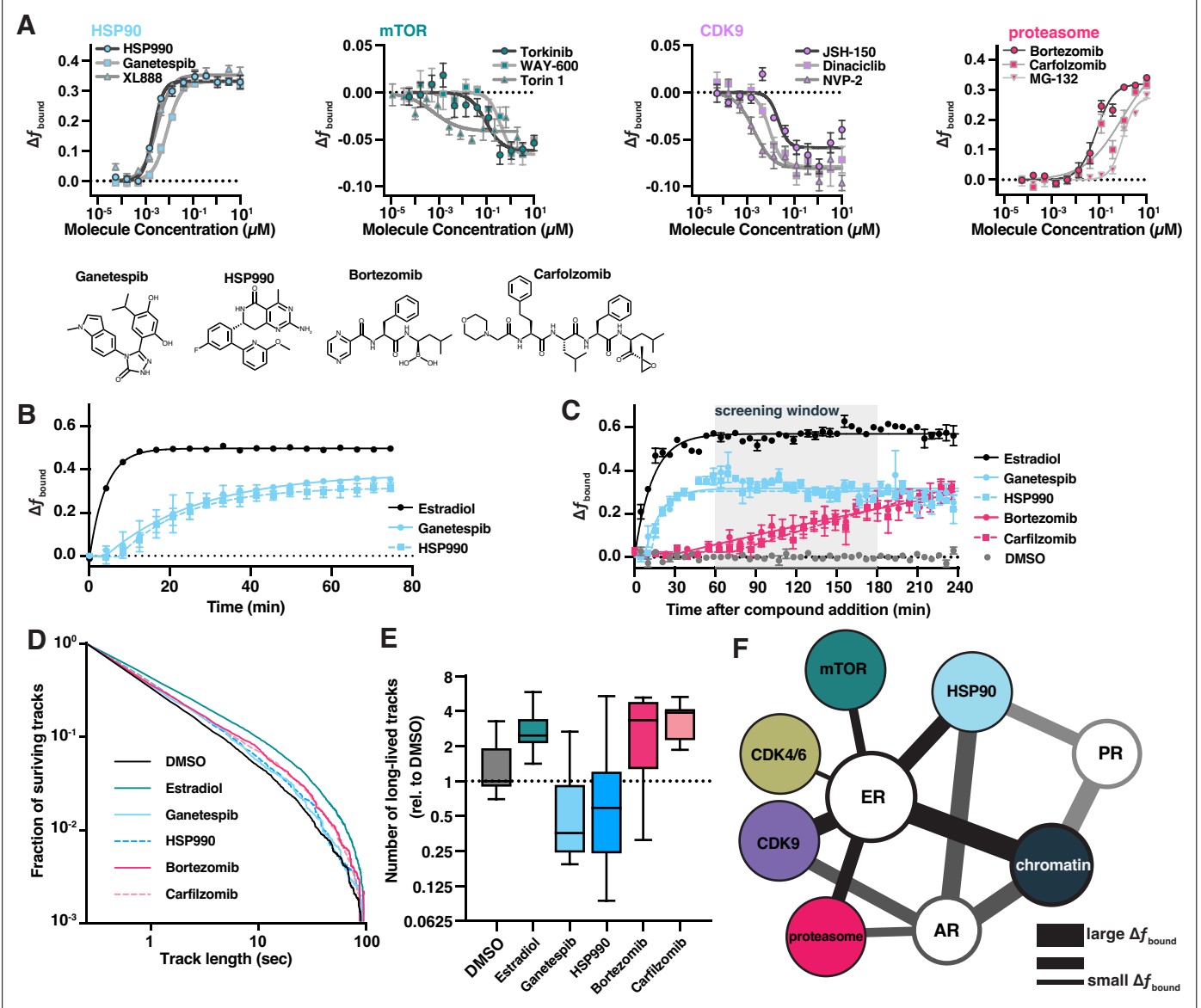

**Figure 6.** Bioactive molecules targeting pathways associated with estrogen receptor (ER) affect its dynamics. (**A**) Bioactive screen results with select inhibitors grouped and uniquely colored by pathway. Change in $f_{bound}$ across a 12-point dose titration of three representative compounds targeting each of HSP90, mTOR, CDK9, and the proteasome. Individual molecules are denoted by specific shapes. Error bars represent SEM across three biological replicates. (**B**) Change in $f_{bound}$ as a function of time after compound addition. Estradiol treatment (black) is compared to ganetespib (blue circles) and HSP990 (blue squares). Points are bins of 4 min. Error bars represent SEM across three biological replicates. (**C**) Change in $f_{bound}$ as a function of time after compound addition. Estradiol treatment (black) is compared to HSP90 inhibitors ganetespib (blue circles) and HSP990 (blue squares); proteasome inhibitors bortezomib (red circles) and carfilzomib (red squares). Points are bins of 7.5 min of high-throughput single-molecule tracking (htSMT) data, marking the mean ± SEM. The shaded region denotes the window of time used during htSMT screening. (**D**) Track length survival curve of ER-Halo cells treated either with DMSO alone, with estradiol stimulation, or with 100 nM HSP90 or proteasome inhibition. Track survival is plotted as the 1-CDF of the track length distribution; faster decay means shorter binding times. (**E**) Quantification of the number of long-lived trajectories as a function of treatment condition. All conditions were normalized to the median number of trajectories in DMSO. (**F**) Diagram summarizing pathway interactions based on htSMT results for ER, AR, and PR.

The online version of this article includes the following source data and figure supplement(s) for figure 6:

**Source data 1.** Tabular data to generate plots from *Figure 6* and related figure supplements.

**Figure supplement 1.** Deeper investigation of bioactive screening data identifies cutoff for active molecules.

**Figure supplement 2.** Some pathway inhibitors modulating estrogen receptor (ER) dynamics are specific to ER.

**Figure supplement 3.** Dose titration plots of ER(S104A/S106A/S118A) with mTOR and CDK9 compounds.

compounds exhibited similar effects based on their biological targets favors the view that these biological targets must themselves interact with ER, and that the compounds therefore affect ER dynamics indirectly. HSP90 is a chaperone for many proteins, including SHRs. In the canonical model, hormone binding releases the SHR–HSP90 complex. Indeed, HSP90 inhibitors increased $f_{bound}$ for ER, AR, and PR, consistent with the hypothesis that one function of the chaperone may be to adjust the equilibrium of SHR binding to chromatin (*Figure 6—figure supplement 2B–D*). Proteasome inhibition also leads to ER immobilization on chromatin (*Stenoien et al., 2001*), which aligns with the results that we obtained in our htSMT screen of bioactive compounds. ER has been shown to be phosphorylated by CDK (*Franco et al., 2018*), Src (*Shah and Rowan, 2005*), or GSK-3 through MAPK and PI3K/AKT signaling pathways (*Anbalagan and Rowan, 2015*), and therefore inhibition of these pathways could reasonably be expected to affect ER dynamics measured using SMT. While CDK inhibition led to an increase in ER mobility, inhibition of PI3K, AKT, or other upstream kinases showed no effect (*Figure 3*).

Interestingly, SMT dynamics of an ER triple point mutant engineered to lack previously defined phosphorylation sites important for transactivation (*Anbalagan and Rowan, 2015*) (S104A/S106A/S118A) were affected by CDK and mTOR pathway inhibitors (*Figure 6—figure supplement 3*), suggesting that additional phosphorylation sites can mediate the effects of CDK9 and PI3K/AKT signaling, or that other molecular targets of CDK and PI3K/AKT can act indirectly to alter the motion of ER. The change in ER protein dynamics for characterized pathway inhibitors such as those targeting CDK and mTOR is subtle but consistent across compounds, suggesting biological meaning in these observations and highlight the need for accurate and precise SMT measurements. Hence, htSMT screening offers the promise of providing comprehensive pathway interaction information or revealing novel interaction mechanisms.

## Kinetic htSMT facilitates evaluation of small molecule mechanism of action

Since SMT can identify compounds that act either directly on a fluorescent target, or through some intermediary process, we sought to distinguish between these alternative modes of action. We hypothesized that by investigating the rate at which changes in protein motion emerge, SMT could be used to distinguish direct versus indirect effects on ER activity. Given the live-cell setting of SMT, we configured a data collection mode that allows for measurement of protein dynamics in set intervals after compound addition (kinetic SMT or kSMT). Both ER agonists and antagonists rapidly induce ER immobilization on chromatin when measured in kSMT ($t_{1/2}$ = 1.6 min for estradiol; *Figure 4C*). On the other hand, HSP90 inhibitors like ganetespib and HSP990 exhibit a delay of 5–7 min before alterations in ER dynamics appear, after which we observed an increase in $f_{bound}$ with a $t_{1/2}$ of 19.3 and 17.5 min, respectively. The overall effect of these compounds reached a plateau after an hour (*Figure 6B*). Proteasome inhibitors, for example bortezomib and carfilzomib, acted even more slowly, with changes in ER dynamics emerging only after 40 min, and slowly increasing over the 4-hr measurement window (*Figure 6C*). Hence, this exploration of SMT kinetics represents an important tool that can facilitate differentiation between on-target and on-pathway modulators. We believe that this approach will permit, for example, rapid mechanistic characterization of active compounds in a drug discovery setting.

To further differentiate the effect of pathway inhibitors on ER protein dynamics, we sought to characterize relative ER residence times for each such molecule. In contrast to the SERDs and SERNs, although HSP90 inhibition by HSP990 and ganetespib resulted in an increase in $f_{bound}$, we observed a decrease in the total number long binding events by two- and fourfold, respectively, while the binding times were similar to that observed with DMSO alone (*Figure 6D, E*). These results suggest HSP90 inhibition primarily increases $k^*_{on}$ while leaving $k^*_{off}$ largely unaffected. On the other hand, inhibition of the proteasome led to an increase in both the number and duration of long binding events. These results demonstrate that ER–chromatin binding can be modulated by changing the rate of association or disassociation, and that the inhibition of specific cellular partners can affect these rates differentially. Taken together with the different kinetics for direct ER, HSP90, and proteasome modulators, our data suggest that each class of molecule alters ER dynamics through distinct mechanisms.

## Discussion

Many pathways that regulate the fundamental biochemistry of cells depend upon the interaction of protein 'sensors' with distinct protein 'effectors' that engage transiently to trigger a change in cell physiology. Although the fundamentals of this process have long been appreciated, biochemical investigation of these protein interactions has typically required in vitro reconstitution or has been interrogated through pull-down assays after cell permeabilization. Here we report the combination of SMT, a type of super-resolution microscopy, with high content microscopy as a means of visualizing individual protein motion in millions of live cells, and under circumstances where the effect of small molecule inhibitors can be assessed quantitatively. We demonstrate the capabilities of the htSMT platform by analyzing the behavior of SHRs, a class of sensors that mediate hormone-induced modulation of gene expression, and in particular the dynamics of the ER.

To validate our htSMT platform, our analysis initially focused on the very rapid immobilization of SHRs on chromatin observed in cells exposed to their established, cognate steroid ligands. The technique proved to be highly quantitative, effectively evaluating ligands whose potency differ by more than four orders of magnitude, and readily characterizing differences in both the sensitivity and the selectivity of the steroid receptor family. These observations prompted us to apply htSMT to screen thousands of bioactive compounds, which we hypothesized would reveal new chemical matter as well as enable comprehensive pharmacologic dissection of ER pathway interactions. As further validation of the technique, automated screening of ER dynamics using htSMT identified all 30 known steroid ligands from a library of 5067 bioactive compounds. The potency of these steroid ligands with respect to alterations in ER dynamics varied across a thousand-fold range, demonstrating the dynamic range of this single experimental setup. Among molecules known to behave as ER signaling inhibitors, the change in ER dynamics correlated closely with the ability of these compounds to block estrogen-induced proliferation of estrogen-dependent breast cancer cells, demonstrating the ability of htSMT to document SARs in a chemical series across disparate and biologically relevant readouts. Since our analysis relies only on detection of changes in protein dynamics, this unbiased readout will prove broadly useful in screening libraries of compounds to identify starting points for the development of new therapeutics. In fact, our analysis identified 209 non-steroidal molecules that affect ER dynamics, which are likely to act elsewhere in the network of ER-interacting proteins.

Our characterization of the htSMT platform and subsequent screen highlighted some important considerations for future screening efforts using SMT. Notably, the observation that cell-to-cell variability is the dominant driver of assay variance, when compared to other sources like the microscope or the assay plate, suggests that an even larger FOV would sample cells more effectively and result in a more stable dynamical estimate (*Figure 3—figure supplement 3*). This could be particularly important for detecting subtle dynamical changes such as those seen with the mTOR inhibitors where the maximal change in $f_{bound}$ was only around 5% (*Figure 6*). Similarly, for our screening assay we chose a frame interval of 10 ms, which proved very sensitive to detecting a wide range of compound effects but is necessarily limited in the types of perturbations it can detect. The diffusion of the most mobile ER population was well below the upper limit of detection for 10 ms, suggesting that faster frame rates were not necessary, but this may not be the case for other protein targets. On the other hand, ER has been reported to have multiple low-mobility chromatin binding states (*Wagh et al., 2023*), but these low-mobility states are below the assay lower bound set by our localization error ($D_{loc} = \frac{\text{localization error}^2}{\Delta\tau}$) and would require a slower frame interval to differentiate.

The observation that by screening large libraries of bioactive compounds for an effect on protein motion, htSMT can define networks of biochemical signaling pathways is a critical outcome of this high-throughput platform. Protein dynamics in the cell are not governed by singular interactions between any two proteins but by biochemical networks within the cell that intersect with the protein under observation. Our work enables construction of a putative interaction network connecting nodes involving different proteins, the identities of which were deduced based on the impact of known inhibitors on the dynamics of steroid receptors (*Figure 6F*). The interaction map derived from unbiased htSMT screening recapitulates many known biological interaction partners of the ER, in a single experimental setup using protein motion as a sole readout. More experiments are necessary to determine which nodes represent direct physical interactions and which occur through intermediaries. To our knowledge, the impact of HSP inhibition in increasing ER–chromatin association has never been described, neither has the link between inhibition post-translation modifying enzymes like the

CDKs or mTOR and ER dynamics ever been described. We believe that identifying additional cellular interaction networks through htSMT will provide an important foundation for the broader understanding of biochemical regulatory mechanisms. For example, CDK4/6 inhibitors are co-administered with SERDs and SERMs to improve therapeutic outcomes (*Liang et al., 2021*; *Wardell et al., 2015*). CDK4/6 inhibition only minimally affects ER dynamics in SMT (*Figure 3*), supporting the view that the combination of CDK4/6 inhibition with ER antagonists functions through a non-redundant ER-independent mechanism (*Finn et al., 2016*).

Most cell-based assays cannot easily be configured for kinetic analysis of treatment effects, as these typically involve fixed-endpoint, aggregate readouts. With such endpoint assays, biochemical feedback and compensatory mechanisms often confound interpretation of the direct effect of a change in treatment conditions. SMT, however, when implemented in a high-throughput system that exhibits consistency over long time intervals, permits kinetic analysis of treatment effects on protein dynamics in real time with high reliability. Such kinetic analyses help to define pathway cascades; in a first instance, they can be used to rapidly identify compounds that likely engage directly with a therapeutic target.

Mechanistically, an increase in $f_{bound}$ provoked by a change in cell conditions suggests that molecular interactions with the protein being analyzed have been stabilized (either made more probable or longer-lasting), while a decrease in $f_{bound}$ suggests the opposite: molecular interactions that have been made less robust. Unexpectedly, we observed both SERDs and SERMs, which antagonize ER, cause an increase in nuclear bound fraction, likely via chromatin binding, in a way that mimics what is seen with traditional ER agonists. We showed that this mechanism is not a commonality among all SHRs, and indeed inhibitors of AR and GR behaved more like the prototypical competitive antagonist. Inhibition of HSP90 also produces a marked increase in ER $f_{bound}$, though these binding events are more transient (*Figure 6E*). Transcription factors are thought to find binding sites through free 3D diffusion, 1D sliding, and transient, non-specific DNA binding (*Liu et al., 2015*; *Boka et al., 2021*; *Paakinaho et al., 2017*; *Mazza et al., 2012*; *Izeddin et al., 2014*). ER antagonists may function by promoting ER binding to non-specific decoy chromatin sites, thus reducing the amount of ER able to activate transcription at ER-responsive genes (*Guan et al., 2019*; *Van Royen et al., 2014*; *Van Royen et al., 2012*). Previous work has also shown that SERDs, and to a lesser extent SERMs, induce an alternate conformation of ER, thereby inhibiting co-factor recruitment (*Guan et al., 2019*). If co-factors like CBP and p300 stabilize ER–chromatin binding, then efficacious inhibitors might exhibit shorter binding times compared with agonist stimulation (*Guan et al., 2019*). The SERMs 4-OHT and GDC-0810 dramatically increase ER–chromatin binding frequency and only modestly increase binding times; fulvestrant and GDC-0927 strongly increase both ER–chromatin binding frequency and residence time. Therefore, htSMT suggests that agonists and antagonists of SHR signaling operate through a previously unappreciated, unified mechanism of chromatin immobilization, meriting further investigation.

Lastly, we note that fast-SMT, as implemented here, can define relationships between the structures of chemical inhibitors and their effects on a fundamental property of protein regulatory elements: their dynamics in living cells. In our hands, these measurements proved far more reliable in predicting the ability of an ER antagonist to block cell proliferation than an ER degradation assay, a conclusion that was only able to be drawn due to the scale of htSMT screening capabilities. Notably, saturable dose responses were observed for ER antagonists at much lower concentrations using htSMT than with ER degradation assays, suggesting that this method will be more sensitive for identifying novel therapeutic agents than with corresponding traditional assay formats. Thus, SMT can identify promising compounds the activity of which might not otherwise be measurable, provide insight into their mechanism(s) of action, and in the native environment of the cell. While ER is a transcription factor, the same principles may apply broadly to regulatory mechanisms of proteins with diverse function. In this context, we conclude that the application of technologies for measuring protein dynamics at scale will prove broadly applicable to the elucidation of biological mechanisms.

## Methods
### Cell lines
The *Homo sapiens* cell lines U2OS (ATCC Cat. No. HTB-96; RRID:CVCL_0042), MCF7 (ATCC Cat. No. HTB-22; RRID:CVCL_0031), T47d (ATCC Cat. No. HTB-133; RRID:CVCL_0553), and SK-BR-3 (ATCC

Cat. No. HTB-30; RRID:CVCL_0033) were grown in Dulbecco's Modified Eagle Medium (DMEM, Cat. No. 1056601, Gibco DMEM, high glucose, GlutaMAX Supplement, Thermo Fisher) supplemented with 10% fetal bovine serum (Cat. No. 16000044, Thermo Fisher) and 1% pen-strep (Cat. No. 15140122, Thermo Fisher) and maintained in a humidified 37°C incubator at 5% $CO_2$ and subcultivated approximately every 2–3 days.

### HaloTag-expressing cell lines

For H2B, CaaX, ER, AR, and PR-HaloTag fusions, mammalian expression vectors containing the fusion gene under the control of a weak L30 promoter and containing a Neomycin resistance marker were transfected into U2OS cells at 70% confluence using FuGENE 6 (Cat. No. E2691, Promega). Transfected cells were selected with G418 (Cat. No. 10131027, Thermo Fisher) at 500 µg/ml, then clonally isolated. Clones expressing the desired fusion gene were determined first by staining with 100 nM $JF_{549}$-HTL (Cat. No. GA1110, Promega) and 50 nM Hoechst 33342 and identifying clones with the expected distribution of $JF_{549}$ signal. Between three and six clones were subsequently tested using SMT conditions for response to a control compound, and the most homogenous clones were subsequently expanded for further testing. Unless otherwise specified, all experiments are with a single, clonally isolated cell line. Because U2OS cells express GR endogenously, HaloTag was inserted right before the stop codon of endogenous NR3C1 via homology-directed repair using CRISPR/Cas9. The HaloTag knock-in was validated by imaging using HTL-$JF_{646}$ staining and through DNA sequencing (*Chu et al., 2023*).

### Western blot

Cells were grown in the same conditions as described previously. $1.5 \times 10^6$ cells were seeded per well in a 6-well plate in DMEM overnight, followed by compound treatment (DMSO or 100 nM fulvestrant) the following day for 24 hr. Cells are lysed in 200 µl 1× Cell Lysis Buffer (catalogue number 9803, Cell Signaling). Protein lysate concentration is then determined using BCA protein assay kit (Catalog number 23225, Pierce BCA Protein Assay Kit) following the manufacturer's instructions. Capillary Western Immunoassay were performed using Jess Protein Simple following the manufacturer's instruction (Protein Pimple, USA). Levels of αER (1:100, RM-9101; RRID:AB_149901) were normalized to loading control β-tubulin (1:100, NC0244815 LI-COR 92642213, Thermo Fisher; RRID:AB_2637092). The peaks were analyzed with the Compass software (Protein Simple, USA).

### RNA-seq

Cells were seeded into 12-well tissue culture treated plates at densities of 250,000 cells (U2OS-WT), 200,000 cells (U2OS-ER), or 300,000 cells (MCF7, SKBR3, T47d) per well. Twenty-four hours later, cells were treated with estradiol at a final concentration of 25 nM for 24 hr. To process cells for total RNA, cells were washed twice with ice-cold phosphate-buffered saline (PBS), lysed with 350 µl Buffer RLT (QIAGEN 79216), scraped off the plate (Fisher 08100241), frozen on dry ice and stored at –20°C. Cell lysates were then thawed, homogenized using QIAshredder columns (QIAGEN 79656), and processed through the QIAGEN RNeasy Micro kit (QIAGEN 74004) using the standard protocol and including the optional on-column DNase digestion step (QIAGEN 79254). All samples had an RNA Integrity Number (RIN) score of 10 by TapeStation (Agilent 5067-5576). RNA sequencing libraries were prepared from total RNA by Novogene (CA). In brief, mRNA was purified from total RNA using poly-T oligo-attached magnetic beads and fragmented. First-strand synthesis was performed using random hexamer primers, second-strand synthesis was performed using dTTP, and libraries were prepared after end repair, A-tailing, adapter ligation, amplification, and purification. Libraries were sequenced on an Illumina NovaSeq with paired 150 cycle reads. For data analysis, paired-end reads were aligned to the hg38 reference genome using Hisat2 v2.0.5, featureCounts v1.5.0-p3 was used to count the number of reads mapped to each gene, and differential expression analysis was performed using DESeq2 (1.20.0).

### SMT sample preparation

Cells were seed on tissue culture treated 384-well glass-bottom plates at 6000 cells per well. Seeded cells were then incubated at 37°C and 5% $CO_2$ to allow adhesion overnight. For all SMT experiments, cells were incubated with 5–100 pM of $JF_{549}$-HTL (Cat. No. GA1110, Promega) and 50 nM Hoechst

33342 for an hour in complete medium. Cells were then washed three times in Dulbecco's Phosphate Buffered Saline (DPBS) and twice in imaging media using an EL406 plate washer, which is fluoroBrite DMEM media (Cat. No. A1896701, Thermo Fisher) supplemented with GlutaMAX (Cat. No. 35050079, Thermo Fisher) and the same serum and antibiotics as growth media. Where appropriate, compounds were serially diluted in an Echo Qualified 384-Well Low Dead Volume Source Microplate (0018544, Beckman Coulter) to generate dose-titration source material. Compounds were administered at a final 1:1000 dilution in cell culture medium. Each dose of a compound has at least 2 replicates per plate and 3 plate replicates, 20 DMSO control wells and 2 no dye control wells were randomized across each plate. Unless otherwise specified, compounds were allowed to incubate for an hour at 37°C prior to image acquisition.

## Image acquisition

Unless otherwise stated, all image acquisition using SMT was performed on a custom-built HILO microscope based on a Nikon Ti2, motorized stage, stage top environmental chamber (OKO labs), quad-band filter cube (Chroma), custom laser launch with 405 nm, and 561 nm wavelengths, coupled to a Nikon TIRF illumination module by fiber optic element (KineFlex HPV-P-3-S-405.640–0.7-APC-P2) delivering >10 and >600 mW of power in a gaussian beam with a FWHM of approximately 250 μm to the back focal plane of the objective. Angle of inclination and beam direction were adjusted by micrometer on the TIRF illuminator and empirically set to maximize and flatten the signal across the camera FOV. Fluorescence emission was passed through a high-speed filter wheel (Finger Lakes Instruments) and collected with a backlit CMOS camera (Prime 95b, Teledyne). Images were acquired with a 60× 1.27 NA water immersion objective (Nikon). Environmental chamber was set to 37°C, 95% humidity, and 5% $CO_2$. For each FOV, 200 SMT frames were collected at a frame rate of 100 Hz, with a 2ms stroboscopic laser pulse. Ten frames of the Hoechst channel were collected at the same frame rate for downstream registration of trajectories to nuclei. Sample focus was maintained using the reflection-based Perfect Focus System to determine the position of the coverglass and apply an empirically determined offset to focus into the sample.

## Experiment design and sample size

All assays were designed with high-throughput screening in mind. Unless otherwise stated, experiments were performed with three biological replicates and within each assay plate having at least three well replicates. In instances where a specific plate or a portion of a plate failed to meet assay quality criteria (e.g. no respons from a positive control), those data point were omitted and, where possible, the assay was repeated. For htSMT experiments, samples were prepared and acquired such that a minimum 20 cells were sampled per assay well, resulting in an minimum of 21,000 detections per well; 60 cells and 63,000 detections minimum per condition per assay plate.

## htSMT image analysis

Image acquisition produced one JF$_{549}$ movie and one Hoechst per FOV. The JF$_{549}$ movie was used to track the motion of individual JF$_{549}$ molecules, while the Hoechst movie was used for nuclear segmentation. Tracking was accomplished in three sequential steps – detection, subpixel localization, and linking – using a combination of existing methods. Briefly, spots were detected using a generalized log likelihood ratio detector to test every 11x11 pixel window using a gaussian kernel with a radius of 1.25 pixels and with a log likelihood detection threshold 14 (*Sergé et al., 2008*). After detection, the estimated position of each emitter was refined to subpixel resolution using fitting (*Levenberg, 1944*; *Marquardt, 1963*; *Laurence and Chromy, 2010*) with an integrated 2D Gaussian spot model (*Smith et al., 2010*) starting from an initial guess afforded by the radial symmetry method (*Parthasarathy, 2012*). Detected spots were linked into trajectories using a custom modification of a hill-climbing algorithm with a maximum linking radius of 1.25 μm and allowing a maximum of two gap frames where a spot may go undetected but still be linked within the same trajectory (*Chenouard et al., 2014*; *Sbalzarini and Koumoutsakos, 2005*). The same detection, subpixel localization, and linking settings were used for all movies used in this manuscript.

For nuclear segmentation, all frames of the Hoechst movie were averaged to generate a mean projection. This mean projection was then segmented with a neural network trained on human-labeled nuclei, the output of which is a mask assigning a semantic category to each pixel in the

image (*Ronneberger et al., 2015*). Each spot was assigned to at most one nucleus using its subpixel coordinates.

To recover dynamical information from trajectories, we used state arrays (*Heckert et al., 2022*), a Bayesian inference approach, with the 'RBME' likelihood function and a grid of 100 diffusion coefficients from 0.01 to 100.0 $\mu m^2\ s^{-1}$ and 31 localization error magnitudes from 0.02 to 0.08 $\mu m$. For each assay well, 10,000 trajectories were randomly sampled from the aggregated pool of nuclear trajectories, and this set of trajectories was used for state inference. After inference, localization error was marginalized out to yield a one-dimensional distribution over the diffusion coefficient for each FOV. For single-cell analysis, we performed SMT and nuclear segmentation on a mixture of U2OS cells bearing H2B-HaloTag.

HaloTag-CaaX, or free HaloTag. We then evaluated the marginal likelihood of each of a set of 100 diffusion coefficients on the set of trajectories within each segmented nucleus (*Quatrini and Ugolini, 2021*). These marginal likelihood functions were clustered with $k$-means (three clusters), and the marginal likelihood functions for each cell were ordered by their cluster index to produce the heatmap. To estimate the fraction bound ($f_{bound}$), we integrated the state array posterior distribution below 0.1 $\mu m^2\ s^{-1}$. To estimate the free diffusion coefficient ($D_{free}$), we computed the mean of the posterior distribution above 0.1 $\mu m^2\ s^{-1}$ using the following equation:

$$D_{\text{free}} = \frac{\sum\limits_{i=25}^{100} D_i * P\left(D_i\right)}{\sum\limits_{i=25}^{100} P\left(D_i\right)}$$

## htSMT data quality control

High content imaging approaches require image quanity control to systematically remove aberrant measurements from the set. Primarily these are fields of view that are empty, that are out of focus, or contain large piece of debris in the well or occlusion of the objective. To detect and remove these fields of view, we trained a convolutional neural network classifier to score Hoechst image quality on a scale of –1 to 1 (low quality to high quality). Annotations from five independent annotators on a training set of ~1000 Hoechst images were used as input for the model. After training, a threshold for filtering was empirically set so as to remove problem FOVs such as those exemplified in (*Figure 1—figure supplement 1c*). Screening wells were only considered if more than two fields of view passed QC, and screening plates were only considered if more than eight control wells could be included for normalization. During screening, compounds with a standard deviation in $f_{bound}$ greater than 0.15 were omitted from analysis and rescreened where possible.

## htSMT data analysis

Tracking results from the automated processing pipeline were analyzed using KNIME or Spotfire (TIBCO). Individual $f_{bound}$ or $D_{free}$ measurements were associated with experimental metadata and aggregated by condition. Change in $f_{bound}$ was calculated as the difference between the $f_{bound}$ of each well and the median $f_{bound}$ of DMSO in the same plate. Wells that had no cells in the FOV or in which the FOV was out of focus were omitted from further analysis. Compounds were assessed for assay interference using the median fluorescence intensity of the tracking channel and omitted if it was more than 3 standard deviations higher than the median intensity of the DMSO wells. Similarly, plates where the active and negative controls could not be clearly resolved or where the significantly deviated from the performance of the rest of the screen were removed from further analysis. Finally, compound with a variance more than three standard deviations higher than the average compound variance (41 compounds; 0.08%) were removed from downstream analysis. $Z'$-factor between the active controls on a plate and DMSO was calculated as previously described (*Zhang et al., 1999*). $EC_{50}$ values were calculated in Prism (GraphPad) by first log-transforming the molecule concentrations and then fitting to a four parameter logistic curve.

## Sources of variability analysis

The contribution of microscope-to-microscope, plate-to-plate, well-to-well, FOV-to-FOV and cell-to-cell variability on the 2D jump distribution was estimated using a subsampling approach. We consider

a simplified model where the observed jump length distribution ($Y$) is a function of the intrinsic stochasticity in jump length due to diffusion ($X$) with additional biases introduced at the cell, FOV, well, plate or microscope level:

$$Y = B_{\mathrm{microscope}} + B_{\mathrm{plate}} + B_{\mathrm{well}} + B_{\mathrm{FOV}} + B_{\mathrm{cell}} + X$$

For simplicity the biases are assumed independent, although these biases may indeed have some dependence.

To estimate the variance of each bias term, we computed the variance in sample means for different resampling schemes. Firstly we sample $N$ jumps from all data from one replicate of the bioactive screen and compute the average. The resulting sample mean is the average over all sources of variability ($B_{\mathrm{microscope}}$, etc.). In a second step, we sample a microscope and randomly sample all jumps from all plates on that microscope. This averages over all other sources of variability except $B_{\mathrm{microscope}}$ can be expected to approach $\mathrm{Var}(B_{\mathrm{microscope}})$ for large $N$. We continue this same sampling scheme at the Plate-, Well-, FOV-, and cell-level, such that for large $N$ results in estimated $\mathrm{Var}(B_{\mathrm{plate}})$, $\mathrm{Var}(B_{\mathrm{well}})$, $\mathrm{Var}(B_{\mathrm{FOV}})$, and $\mathrm{Var}(B_{\mathrm{cell}})$, respectively. 1000 rounds of sampling were performed for each resampling scheme, either including all wells or only those containing the vehicle DMSO.

## Clustering active molecules

Chemical structure-based clustering was performed on molecules identified as active (239 in total). Molecular frameworks were computed as described by Murcko et al. and as implemented in Pipeline Pilot (*Bemis and Murcko, 1996*). Molecular frameworks were clustered using functional class fingerprints (FCFP_4) with a similarity threshold cut-off of 0.3 Tanimoto distance. A total of 21 clusters were obtained with singletons being the major class (124 molecules). The next largest group was the flavone class represented by 27 members, followed by a couple of diverse classes within the steroidal class with 14 and 20 members, respectively. The other category is the stilbene class with seven members representing tamoxifen as one of the members. The remaining actives (47 molecules) were grouped into one 3-membered cluster and all the others with 2 members per cluster.

## Kinetic experiments

Cells were seeded into a 384-well plate the day before, dyed, and washed as described above. 1 well with 25 FOVs per well were taken as a baseline reading. Then, while imaging, compound was manually added to each well to a final concentration of 100 nM. Data were then collected for 20 wells. A pause was included between each FOV such that the entire imaging regime covers the assay window. Change in $f_{\mathrm{bound}}$ was determined per-well relative to $t = 0$.

For assays extending to 4 hr, the plate was imaged twice with 8 FOVs per well with different FOV locations per readthrough to prevent photobleaching from impacting data. All data presented represents was performed in three different biological replicates.

## Residence time imaging

Sample preparation and execution of residence time imaging experiments were conducted in a similar manner to the SMT assay described above with a few exceptions. Samples were dyed with 1–10 pM JF$_{549}$ (Promega) and 50 nM Hoechst 33342 for an hour. 400 frames per FOV were collected with a camera integration time was set to 250 ms, and laser sources reduced to 5 mW at the objective. During image acquisition, lasers were on continuously. Compound incubation ranged from 1 to 4 hr. At least 8 well replicates were collected per condition.

## Residence time analysis

Quantifying transcription factor binding times on DNA is an open problem with multiple proposed solutions (*Mazza et al., 2012*; *Garcia et al., 2021*; *Reisser et al., 2020*). Here, we adopted an approach similar to *Hansen et al., 2017*. Image processing, including spot detection, localization, and track reconnection were performed using the same methods described above. Because residence time imaging selectively tracks slow-diffusing molecules, individual localizations were limited to a 300-nm maximum displacement for individual jump reconnections. Sets of trajectories for each FOV were binned into 1-CDF distributions as previously described and fit to a two exponent decay model:

$$\text{CDF}\,(t) = A\left(Fe^{-k_{\text{fast}}t} + (1-F)\,e^{-k_{\text{slow}}t}\right)$$

The $k_{\text{slow}}$ term comprises both the rate of molecule unbinding ($k_{\text{off}}$) as well as photobleaching ($k_{\text{bleach}}$) and diffusion of chromatin out of the focal volume. Often approaches attempt to derive the unbinding rate by applying a correction such as subtracting a bleaching rate measured either directly in the sample or by using a separate control sample (*Hansen et al., 2017*; *Mazza et al., 2012*), where $T_{\text{corrected}} = (k_{\text{off}} - k_{\text{bleach}})^{-1}$. The inverse relationship between $T_{\text{corrected}}$ and $k_{\text{bleach}}$ makes it highly sensitive and nonlinear to noise in $k_{\text{bleach}}$. Because the 1-CDF distributions of compound-treated ER samples are so close in decay rate to the control Histone H2B, background subtraction yields unfeasible $k_{\text{off}}$ values after correction. Instead we report only the uncorrected $k_{\text{slow}}$ values with the understanding that these represent a lower-bound of the actual $k_{\text{off}}$, but that within-experiment comparisons can be made between conditions.

## Fluorescence recovery after photobleaching

Images were acquired on a custom-built HiLo microscope as described above with a Spectra Light Engine RS-232. Stimulation was directed using a miniscanner coupled with a Coherent OBIS 561 nm 100 mW laser. All imaging was performed using a 60× 1.27 NA water immersion objective (Nikon). All experiments were performed at 37°C. For FRAP experiments, cells were seeded into a 384-well plate the day before, labeled with 50 nM HTL-JF$_{549}$, and washed as described above. Compound was added to 100 nM final an hour before imaging. Then, a prebleach image was acquired by averaging 10 consecutive images. Then 8–10 regions were bleached (2 background, 6–8 cells) and 2 regions in cells were unbleached. Regions that were bleached were bleached at 10% power without scanning. For the next 30 s, an image was acquired every 200 ms, then every 1 s for 2 min. The background-subtracted average intensity was measured in the region of interest over time and normalized to the average of the fluorescence in the baseline images, then normalized to the unbleached regions to account for readout-induced photobleaching of fluorophores. Data from 18 to 24 cells were pooled per experiment for 3 biological experiments.

## Immunofluorescence

Cells were grown in conditions as described previously. Cells were seeded in glass bottom 384-well plates coated with 0.05 mg/ml PDL (Cat. No. A3890401, Thermo Fisher) at 6000 cells per well for Halo-ER U2OS cells and 8000 for MCF7 and T47d cells. Cells were grown overnight followed by compound treatment on the second day for 24 hr at 37°C and 5% CO$_2$. Compounds were serially diluted in an Echo Qualified 384-Well Low Dead Volume Source Microplate (0018544, Beckman Coulter) to generate a 21-point dose response at 1:3 dilution starting from a concentration of 10 mM. Compounds were administered at a final 1:1000 dilution in cell culture medium. An 8- to 12-point dose response was selected based on the potency of each compound. Each concentration was replicated at least once per plate and has at least two plate replicates. Cells were fixed by addition of paraformaldehyde (Cat. No. 15710-S; Electron Microscopy Sciences), with a final concentration of 4% for 20 min. Cells were then permeabilized using blocking buffer containing 1% bovine serum albumin and 0.3% Triton X-100 in 1× PBS for an hour at room temperature. Immunofluorescent staining of ER was carried out using αER antibody (1:500, RM-9101; RRID:AB_149899) diluted in the same blocking buffer for 1 hr at room temperature. Extensive washing with PBS was performed prior to secondary antibody staining. Secondary antibody staining was carried out using Alexa fluor 488 conjugate anti-rabbit IgG (1:1000, Cat. No. A32731, Thermo Fisher; RRID:AB_2633280) for an hour. Nuclear staining was carried out using Hoechst 33342 solution at 1 mg/ml. Imaging of immunofluorescence was done using the ImageXpress Micro (Molecular Devices) at ×10 magnification and 4 FOV per well. Fluorescence intensity within the nucleus was quantified using CellProfiler (*Stirling et al., 2021*). All analysis and curve fitting were carried out using Prism with DMSO as a baseline. ER degradation experiments were performed in three biological replicates with the same source compounds.

## Cell proliferation

Cells were grown and seeded in conditions as described above. Cells were seeded in 384well plates (Cat. No. 353963, Corning) at 1000 cells per well for Halo-ER U2OS, 1200 cells for SKBR3, and 1800 cells for MCF7 and T47d. Cells were grown overnight, then treated with compounds the

following day. Compound concentration and administration are the same as described previously for the immunofluorescence assay. Plates are scanned in the IncuCyte live cell analysis system (Sartorius) at 24-hr intervals for a total of 5 days using phase contrast. Cell proliferation quantification was carried out by the built-in analysis function using whole well confluency mask. All analysis and curve fitting were carried out using Prism with DMSO as a baseline. MCF7 and T47d cell proliferation experiments were performed in four biological replicates with the same source compounds.

## Acknowledgements

The authors extend their deepest gratitude to all the employees and consultants of Eikon, past and present, especially Caitlyn Bonilla, Tiffany Cheng, Michael Hirte, David Hoffman, Fedor Ilkov, Yan Li, Anuja Lohia, Edith Martinez-Soto, Eugene Masterov, Mai Nguyen, and Gregory Snyder. Their tireless work enabled the experiments described here. We thank Rand Miller, Roger Perlmutter, Yan Li, Robert Tjian for helpful discussions and critical feedback on the direction of our investigation and on the resulting manuscript. Eikon Therapeutics provided all funding. No external funding was received for this work.

## Additional information

### Competing interests

David Trombley McSwiggen, Helen Liu, Ruensern Tan, Sebastia Agramunt Puig, Lakshmi B Akella, Russell Berman, Mason Bretan, Hanzhe Chen, Xavier Darzacq, Kelsey Ford, Ruth Godbey, Eric Gonzalez, Adi Hanuka, Alec Heckert, Jaclyn J Ho, Stephanie L Johnson, Reed Kelso, Aaron Klammer, Ruchira Krishnamurthy, Jifu Li, Kevin Lin, Brian Margolin, Patrick McNamara, Laurence Meyer, Sarah E Pierce, Akshay Sule, Connor Stashko, Yangzhong Tang, Daniel J Anderson, Hilary P Beck: Employee of Eikon Therapeutics Inc.

### Funding

No external funding was received for this work.

### Author contributions

David Trombley McSwiggen, Conceptualization, Resources, Data curation, Formal analysis, Supervision, Investigation, Visualization, Methodology, Writing – original draft, Writing – review and editing; Helen Liu, Ruensern Tan, Data curation, Investigation, Visualization, Methodology, Writing – review and editing; Sebastia Agramunt Puig, Adi Hanuka, Software, Visualization; Lakshmi B Akella, Data curation, Visualization; Russell Berman, Conceptualization, Software, Supervision, Methodology, Writing – review and editing; Mason Bretan, Software, Visualization, Methodology; Hanzhe Chen, Resources, Investigation, Methodology; Xavier Darzacq, Conceptualization, Writing – review and editing; Kelsey Ford, Akshay Sule, Resources, Investigation; Ruth Godbey, Jifu Li, Software; Eric Gonzalez, Resources, Methodology; Alec Heckert, Software, Formal analysis, Visualization, Methodology, Writing – review and editing; Jaclyn J Ho, Supervision, Methodology, Writing – review and editing; Stephanie L Johnson, Software, Supervision; Reed Kelso, Aaron Klammer, Software, Supervision, Methodology; Ruchira Krishnamurthy, Investigation; Kevin Lin, Brian Margolin, Supervision, Methodology; Patrick McNamara, Laurence Meyer, Software, Methodology; Sarah E Pierce, Formal analysis, Investigation, Methodology; Connor Stashko, Formal analysis; Yangzhong Tang, Resources, Supervision; Daniel J Anderson, Conceptualization, Supervision, Writing – original draft, Writing – review and editing; Hilary P Beck, Conceptualization, Data curation, Supervision, Methodology, Writing – original draft, Writing – review and editing

### Author ORCIDs
Xavier Darzacq ⓘ https://orcid.org/0000-0003-2537-8395
Brian Margolin ⓘ https://orcid.org/0000-0003-3365-7677
Patrick McNamara ⓘ https://orcid.org/0000-0003-2756-0887
Hilary P Beck ⓘ https://orcid.org/0000-0002-5003-1361

Reviewer #1 (Public review): https://doi.org/10.7554/eLife.93183.3.sa1
Reviewer #3 (Public review): https://doi.org/10.7554/eLife.93183.3.sa2
Author response https://doi.org/10.7554/eLife.93183.3.sa3

## Additional files

### Supplementary files
MDAR checklist

### Data availability
The process of generating single-molecule tracking data as implemented in this manuscript is both data storage and data processing intensive. The data in this manuscript alone is ~100 terabytes, and there is no database in existence appropriate to host or make available these data in a practical manner. Similarly, the computational resources necessary to process the full datasets would be costly and impractical for labs wishing to process the entire dataset. The core of the image processing and tracking algorithms are implementations of the works we cite in our methods section, but the specific software written for the purposes of processing and analyzing these large image sets was developed with the specific compute and storage infrastructure of Eikon in mind. In a subsequent manuscript, we have provided an example of how the processing pipeline functions operate (*Driouchi et al., 2023*). Where possible we have provided the numerical tables of processed data sufficient to reproduce the plots and figures. In addition, we have provided example files with the inputs (raw image files) and outputs of segmentation, particle tracks, and state array distributions which can be found in the following data repository: https://doi.org/10.5061/dryad.xd2547dsw. Parties interested in accessing any additional data contained within the manuscript should contact the corresponding author to determine which non-proprietary data are needed and in what format they will be provided through a Data Use Agreement with Eikon Therapeutics. Other materials will be made available through a Materials Transfer Agreement with Eikon Therapeutics by contacting the corresponding author.

The following dataset was generated:

| Author(s) | Year | Dataset title | Dataset URL | Database and Identifier |
|---|---|---|---|---|
| McSwiggen DT | 2024 | Example single molecule tracking data from "A high-throughput platform for single-molecule tracking identifies drug interaction and cellular mechanisms" | https://doi.org/10.5061/dryad.xd2547dsw | Dryad Digital Repository, 10.5061/dryad.xd2547dsw |

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
