## [Editor Report · eLife assessment]

This work presents an **important** technological advance, in the form of a high throughput platform for Single Particle Tracking allowing us to measure millions of cells and thousands of compounds per day. Analysis of the diffusional behaviour of fluorescently-tagged targets permits the identification of, and differentiation between, small molecules that bind directly or affect the target indirectly. The methodology and metrics employed are **compelling**, leading to the identification of multiple compounds that effectively change the diffusive state of the estrogen receptor, the POC target of the study.

---

## [Referee Report · Reviewer #1 (Public review)]

Summary:

The authors set up a pipeline for automated high-through single-molecule fluorescence imaging (htSMT) in living cells and analysis of molecular dynamics.

Strengths:

htSMT reveals information on the diffusion and bound fraction of molecules, dose-response curves, relative estimates on binding rates, and temporal changes of parameters. It enables the screening of thousands of compounds in a reasonable time and proves to be more sensitive and faster than classical cell-growth assays. If the function of a compound is coupled to the mobility of the protein of interest or affects an interaction partner, which modulates the mobility of the protein of interest, htSMT allows identifying the modulator and getting the first indication on the mechanism of action or interaction networks, which can be a starting point for more in-depth analysis. The authors describe their automated imaging and analysis procedures as well as the measures taken to assure data and analysis quality.

Weaknesses:

While elegantly showcasing the power of high-throughput measurements, htSMT relies on a sophisticated robot-based workflow and several microscopes for parallel imaging, thus limiting wide-spread application of htSMT by other scientists.

---

## [Referee Report · Reviewer #3 (Public review)]

Summary:

The authors aim to demonstrate the effectiveness of their developed methodology, which utilizes super-resolution microscopy and single-molecule tracking in live cells on a high-throughput scale. Their study focuses on measuring the diffusion state of a molecule target, the estrogen receptor, in both ligand-bound and unbound forms in live cells. By showcasing the ability to screen 5067 compounds and measure the diffusive state of the estrogen receptor for each compound in live cells, they illustrate the capability and power of their methodology.

Readers are well introduced to the principles in the initial stages of the manuscript with highly convincing video examples. The methods and metrics used (fbound) are robust. The authors demonstrate high reproducibility of their screening method (R2=0.92). They also showcase the great sensitivity of their method in predicting the proliferation/viability state of cells (R2=0.84). The outcome of the screen is sound, with multiple compounds clustering identified in line with known estrogen receptor biology.

---

## [Author Response]

The following is the authors’ response to the original reviews.

**Reviewer #1 (Public Review):**
Summary:The authors set up a pipeline for automated high-throughput single-molecule fluorescence imaging (htSMT) in living cells and analysis of molecular dynamicsStrengths:htSMT reveals information on the diffusion and bound fraction of molecules, dose-response curves, relative estimates of binding rates, and temporal changes of parameters. It enables the screening of thousands of compounds in a reasonable time and proves to be more sensitive and faster than classical cell-growth assays. If the function of a compound is coupled to the mobility of the protein of interest, or affects an interaction partner, which modulates the mobility of the protein of interest, htSMT allows identifying the modulator and getting the first indication of the mechanism of action or interaction networks, which can be a starting point for more in-depth analysis.Weaknesses:While elegantly showcasing the power of high-throughput measurements, the authors disclose little information on their microscope setup and analysis procedures. Thus, reproduction by other scientists is limited. Moreover, a critical discussion about the limits of the approach in determining dynamic parameters, the mechanism of action of compounds, and network reconstruction for the protein of interest is missing. In addition, automated imaging and analysis procedures require implementing sensitive measures to assure data and analysis quality, but a description of such measures is missing.

The reviewer rightly highlights both the power and complexity in high throughput assay systems, and as such the authors have spent significant effort in first developing quality control checks to support screening. We discuss some of these as part of the description and characterization of the platform. We added additional details into the manuscript to help clarify. The implementation of our workflow for image acquisition, processing and analysis relies heavily on the specifics of our lab hardware and software infrastructure. We have added additional details to the text, particularly in the Methods section, and believe we have added enough information that our results can be reproduced using the suite of tools that already exist for single molecule tracking.

The reviewer also points out that all assays have limitations, and these have not been clearly identified as part of our discussion of the htSMT platform. We have also added some comments on the limitations of the current system and our approach.

**Reviewer #2 (Public Review):**
Summary:McSwiggen et al present a high throughput platform for SPT that allows them to identify pharmaceutics interactions with the diffusional behavior of receptors and in turn to identify potent new ligands and cellular mechanisms. The manuscript is well written, it provides a solid new mentor and a proper experimental foundationStrengths:The method capitalizes and extends to existing high throughput toolboxes and is directly applied to multiple receptors and ligands. The outcomes are important and relevant for society. 10^6 cells and >400 ligands per is a significant achievement.The method can detect functionally relevant changes in transcription factor dynamics and accurately differentiate the ligand/target specificity directly within the cellular environment. This will be instrumental in screening libraries of compounds to identify starting points for the development of new therapeutics. Identifying hitherto unknown networks of biochemical signaling pathways will propel the field of single-particle live cell and quantitative microscopy in the area of diagnostics. The manuscript is well-written and clearly conveys its message.Weaknesses:There are a few elements, that if rectified would improve the claims of the manuscript.The authors claim that they measure receptor dynamics. In essence, their readout is a variation in diffusional behavior that correlates to ligand binding. While ligand binding can result in altered dynamics or /and shift in conformational equilibrium, SPT is not recording directly protein structural dynamics, but their effect on diffusion. They should correct and elaborate on this.

This is an excellent clarifying question, and we have tried to make it more explicit in the text. The reviewer is absolutely correct; we’re not using SPT to directly measure protein structural dynamics, but rather the interactions a given protein makes with other macromolecules within the cell. So when an SHR binds to ligand it adopts conformations that promote association with DNA and other protein-protein interactions relevant to transcription. This is distinct from assays that directly measure conformational changes of the protein.

L 148 What do the authors mean 'No correlation between diffusion and monomeric protein size was observed, highlighting the differences between cellular protein dynamics versus purified systems'. This is not justified by data here or literature reference. How do the authors know these are individual molecules? Intensity distributions or single bleaching steps should be presented.

The point we were trying to make is that the relative molecular weights for the monomer protein (138 kDa for Halo-AR, 102 kDa for ER-Halo, 122 kDa for Halo-GR, and 135 kDa for Halo-PR) is uncorrelated with its apparent free diffusion coefficient. Were we to make this measurement on purified protein in buffer, where diffusion is well described by the Stokes Einstein equation, one would expect to see monomer size and diffusion related. We’ve clarified this point in the manuscript.

Along the same lines, the data in Figs 2 and 4 show that not only the immobile fraction is increased but also that the diffusion coefficient of the fast-moving (attributed to free) is reduced. The authors mention this and show an extended Fig 5 but do not provide an explanation.

This is an area where there is still more work to do in understanding the estrogen receptor and other SHRs. As the reviewer says, we see not only an increase in chromatin binding but also a decrease in the diffusion coefficient of the “free” population. A potential explanation is that this is a greater prevalence of freely-diffusing homodimers of the receptor, or other protein-protein interactions (14-3-3, P300, CBP, etc) that can occur after ligand binding. Nothing in our bioactive compound screen shed light on this in particular, and so we can only speculate and have refrained from drawing further conclusions in the text.

How do potential transient ligand binding and the time-dependent heterogeneity in motion (see comment above) contribute to this? Also, in line 216 the authors write "with no evidence" of transient diffusive states. How do they define transient diffusive states? While there are toolboxes to directly extract the existence and abundance of these either by HMM analysis or temporal segmentation, the authors do not discuss or use them.

Throughout the analysis in this work, we consider all of tracks with a 2-second FOV as representative of a single underlying population and have not looked at changes in dynamics within a single movie. As we show in the supplemental figures we added (see Figure 3, figure supplement 1), this appears to be a reasonable assumption, at least in the cases we’ve encountered in this manuscript. For experiments involving changes in dynamics over time, these are experiments where we’ve added compound simultaneous with imaging and collect many 2-second FOVs in sequence to monitor changes in ER dynamics. In this case when we refer to “transient states,” we are pointing out that we don’t observe any new states in the State Array diagram that exist in early time points but disappear at later time point.

The reviewer suggests track-level analysis methods like hidden Markov models or variational Bayesian approaches which have been used previously in the single molecule community. These are very powerful techniques, provided the trajectories are long (typically 100s of frames). In the case of molecules that diffuse quickly and can diffuse out of the focal plane, we don’t have the luxury of such long trajectories. This was demonstrated previously (Hansen et al 2017, Heckert el al 2022) and so we’ve adopted the State Array approach to inferring state occupations from short trajectories. As the reviewer rightly points out, this approach potentially loses information about state transitions or changes over time, but as of now we are not aware of any robust methods that work on short trajectories.

The authors discuss the methods for extracting kinetic information of ligand binding by diffusion. They should consider the temporal segmentation of heterogenous diffusion. There are numerous methods published in journals or BioRxiv based on analytical or deep learning tools to perform temporal segmentation. This could elevate their analysis of Kon and Koff.

We’re aware of a number of approaches for analyzing both high framerate SMT as well as long exposure residence time imaging. As we say above, we’re not aware of any methods that have been demonstrated to work robustly on short trajectories aside from the approaches we’ve taken. Similarly, for residence time imaging there are published approaches, but we’re not aware of any that would offer new insight into the experiments in this study. If the reviewer has specific suggestions for analytical approaches that we’re not aware of we would happily consider them.

**Reviewer #3 (Public Review):**
Summary:The authors aim to demonstrate the effectiveness of their developed methodology, which utilizes super-resolution microscopy and single-molecule tracking in live cells on a high-throughput scale. Their study focuses on measuring the diffusion state of a molecule target, the estrogen receptor, in both ligand-bound and unbound forms in live cells. By showcasing the ability to screen 5067 compounds and measure the diffusive state of the estrogen receptor for each compound in live cells, they illustrate the capability and power of their methodology.Strengths:Readers are well introduced to the principles in the initial stages of the manuscript with highly convincing video examples. The methods and metrics used (fbound) are robust. The authors demonstrate high reproducibility of their screening method (R2=0.92). They also showcase the great sensitivity of their method in predicting the proliferation/viability state of cells (R2=0.84). The outcome of the screen is sound, with multiple compounds clustering identified in line with known estrogen receptor biology.Weaknesses:Potential overstatement on the relationship of low diffusion state of ER bound to compound and chromatin state without any work on chromatin level.

We appreciate the reviewers caution in over-interpreting the relationship between an increase in the slowest diffusing states that we observe by SMT and bona fide engagement with chromatin. In the case of the estrogen receptor there is strong precedent in the literature showing increases in chromatin binding and chromatin accessibility (as measured by ChIP-seq and ATAC-seq) upon treatment with either estradiol or SERM/Ds. Taken together with the RNA-seq, we felt it reasonable to assume all the trajectories with a diffusion coefficient less that 0.1 µm2/sec were chromatin bound.

Could the authors clarify if the identified lead compound effects are novel at any level?

Most of the compounds we characterize in the manuscript have not previously been tested in an SMT assay, but many are known to functionally impact the ER or other SHRs based on other biochemical and functional assays. We have not described here any completely novel ER-interacting compounds, but to our knowledge this is the first systematic investigation of a protein showing that both direct and indirect perturbation can be inferred by observing the protein’s motion. Especially for the HSP90 inhibitors, the observation that inhibiting this complex would so dramatically increase ER chromatin-binding as opposed to increasing the speed of the free population is counterintuitive and novel.

More video example cases on the final lead compounds identified would be a good addition to the current data package.
**Reviewer #1 (Recommendations For The Authors):**
General:More information on the microscope setup and analysis procedures should be given. Since custom code is used for automated image registration, spot detection, tracking, and analysis of dynamics, this code should be made publicly available.Results:line 97: more details about the robotic system and automatic imaging, imaging modalities, and data analysis procedures should be given directly in the text.

Additional information added to text and methods

line 100: we generated three U2OS cell lines  how?

Additional information added to text and methods

line 101: ectopically expressing HaloTag fused proteins  how much overexpression did cells show?

The L30 promoter tends to produce fairly low expression levels. The same approach was used for all ectopic expression plasmids, and for the SHRs the expression levels were all comparable to endogenous levels. We have not checked this for H2B, Caax and free Halo but given that the necessary dye concentration to achieve similar spot densities is within a 10-fold range for all constructs, its reasonable to say that those clonal cell lines will also have modest Halotag expression.

line 107: Single-molecule trajectories measured in these cell lines yielded the expected diffusion coefficients  how was data analysis performed?

Additional information added to text and methods

line 109: how was the localization error determined?

Additional information added to text and methods

line 155: define occupation-weighted average diffusion coefficient.

Additional information added to text and methods

line 157: with 34% bound in basal conditions and 87% bound after estradiol treatment contradicts figure 2b, where the bound fraction is up to 50% after estradiol treatment.

Line 157 is the absolute fraction bound, figure 2b is change in fbound

line 205: Figure 2c is missing.

Fixed

line 215: within minutes  how was this data set obtained? which time bins were taken?

Additional information added to text and methods

line 216: with no evidence of transient diffusive states What is meant by transient diffusive state? It seems all time points have a diffusive component, which decreases over time.

Additional information added to text and methods

The diffusive peak decreases, the bound peak increases but no other peaks emerge during that time (e.g. neither super fast nor super slow)

line 225: it seems that fbound of GDC-0810 and GDC-0927 are rather similar in FRAP experiments, please comment, how was FRAP done?

FRAP is in the methods section. The curves and recovery times are quite distinct, is the reviewer looking at

line 285: reproducibly: how often was this repeated?

Information added to the manuscript

line 285: it would be necessary to name all of the compounds that were tested, e.g. with an ID number in the graph and a table. This also refers to extended data 7 and 8.

Additional supplemental file with the list of bioactive compounds tested will be included.

line 290/1: what is meant by vendor-provided annotation was poorly defined?

Additional information added to text and methods. Specifically, the “other” category is the most common category, and it includes both compounds with unknown targets/functions as well as compound where the target and pathway are reasonably well documented. Hence, we applied our own analysis to better understand the list of active compounds.

Figures:fig. 2-6: detailed statistics are missing (number of measured cells, repetitions, etc.).

We have added clarifying information, including an “experiment design and sample size” section in the Methods.

fig. 3: the authors need to give a list with details about the 5067 compounds tested,

Additional supplemental file with the list of bioactive compounds tested will be included.

extended data 1c: time axis does not correspond to the 1.5s of imaging in the text, results line 127.

Axes fixed

extended data 3: panel c and d are mislabeled.

Panel labels fixed

Methods:line 746: HILO microscope: the authors need to explain how they can get such large fields of view using HILO

Additional details added to the materials and methods. The combination of the power of the lasers, the size of the incident beam out of the fiber optic coupling device and the sCMOS camera are the biggest components that enable detection over a larger field of view.

line 761: it is common practice to publish the analysis code. Since the authors wrote their own code, they should publish it

Our software contains proprietary information that we cannot yet release publicly. Comparable results can be achieved with HILO data using publicly-available tools like utrack. State Arrays code is distributed and the parameters used are listed in the M&M.

**Reviewer #2 (Recommendations For The Authors):**
The writing and presentation are coherent, concise, and easy to follow.The authors should consider justifying the following:Why is 1.5s imaging time selected? Topological and ligand variations may last significantly longer than this. The authors should present at least for one condition the same effect images for longer.

Related to the similar comment above, we added a figure examining the jump length distribution as a function of frame. Over the 6 seconds of data collection the jump length distribution is unchanged, suggesting it is reasonable to consider all the trajectories within an FOV as representative of the same underlying dynamical states.

The authors miss the k test or T test in their graphs.

We chose to apply the Kurskal-Wallis test in the context of the bioactive screen to assess whether a grouping of compounds based on their presumed cellular target was significantly different from the control even when individual compounds might not by themselves raise to significance. In this case many of the pathway inhibitors are subtle and not necessarily obvious in their difference. In the other cases throughout the manuscript, whether two conditions are statistically distinguishable is rarely in question and of far less importance to the conclusions in the manuscript than the magnitude of the difference. We’ve added statistical tests where appropriate.

The overall integrated area of Fig 4a appears to reduce upon ligand addition. Data appear normalized but the authors should also add N (number of molecules) on top of the graphs.

While the integrated area may appear to decrease, all State Array analysis is performed by first randomly sampling 10,000 trajectories from the assay well and inferring state distribution on those 10,000. This has been clarified in the figure legend and in the Methods.

MinorExtended Figure 3 legend c, d appear swapped and incorrectly named in the text.

Panel labels fixed

L 197 but this appears not to BE a general feature of SHRs (maybe missing Be).

Error fixed

L205 authors refer to Figure 2c, which does not exist.

Panel reference fixed

**Reviewer #3 (Recommendations For The Authors):**
Among minor issues:In Figure 1B, if the authors could specify how they discriminate the specific cell lines from the mixed context, it would enhance clarity. Could they perform additional immunofluorescence to understand how the assignment is determined? Alternatively, could they also show the case with isolated cell lines in an unmixed context?

Immunofluorescence would be a challenge given that there is not a good epitope to distinguish the three ectopically-expressed genes from each other or from endogenous proteins in the case of H2B and CaaX. We are really reliant on the single cell dynamics to determine the likely cell identity. That said, we’ve added graphs of a number of individual cell State Arrays from the same data graphed in 1A which support the notion that it’s reasonable to assume a cells identity given the observed dynamics.

In Extended Figure 2F: possibly a CHip-Seq experiment would be more directly qualified to state the effect of ER ligand on ER ability to bind chromatin.

This is true. Presumably ER that is competent at activating transcription of ER-responsive genes is also capable of binding DNA. ChIP would be the more direct measure, but would not address whether the protein was functional. We chose to balance these measuring these two aspects of ER biology by pairing dynamics with the end-point transcription readout.

In Figure 3: A representation with plate-by-plate orientation along the x-axis, with controls included in each plate, would be more appropriate to reflect the consistency of the controls used in the assay across different plates. Currently, all controls are pooled in one location, and we cannot appreciate how the controls vary from plate to plate.

Figure added to the supplement

Also in this figure, a general workflow of the screen down to segmentation/analysis would be a great add-on.

New figure added to the supplement and reflected in the textual description of the platform

In Extended Figures 3B and C an add-on of the positive and negative control would make the figure more convincing.

Addressed as part of figure added to the supplement

Is there any description of compound leads identified that is novel in nature in relation to impact on ER, and if so could it be stated more clearly in the text as novel finding?

To our knowledge, the impact of HSP inhibition in increasing ER-chromatin association has never been described, neither has the link between inhibition post-translation modifying enzymes like the CDKs or mTOR and ER dynamics ever been described. We added clarifying text to the manuscript